# Spectral Graph Pruning Against Over-Squashing and Over-Smoothing

**Adarsh Jamadandi**[*1,2]
adarsh.jam@gmail.com

**Celia Rubio-Madrigal**[*2]
celia.rubio-madrigal@cispa.de

**Rebekka Burkholz**[2]
burkholz@cispa.de

[1] Universität des Saarlandes
[2] CISPA Helmholtz Center for Information Security

## Abstract

Message Passing Graph Neural Networks are known to suffer from two problems that are sometimes believed to be diametrically opposed: *over-squashing* and *over-smoothing*. The former results from topological bottlenecks that hamper the information flow from distant nodes and are mitigated by spectral gap maximization, primarily, by means of edge additions. However, such additions often promote over-smoothing that renders nodes of different classes less distinguishable. Inspired by the Braess phenomenon, we argue that deleting edges can address over-squashing and over-smoothing simultaneously. This insight explains how edge deletions can improve generalization, thus connecting spectral gap optimization to a seemingly disconnected objective of reducing computational resources by pruning graphs for lottery tickets. To this end, we propose a computationally effective spectral gap optimization framework to add or delete edges and demonstrate its effectiveness on the long range graph benchmark and on larger heterophilous datasets.

## 1 Introduction

Graphs are ubiquitous data structures that can model data from diverse fields ranging from chemistry (Reiser et al., 2022), biology (Bongini et al., 2023) to even high-energy physics (Shlomi et al., 2021). This has led to the development of deep learning techniques for graphs, commonly referred to as Graph Neural Networks (GNNs). The most popular GNNs follow the message-passing paradigm (Gori et al., 2005; Scarselli et al., 2009; Gilmer et al., 2017; Bronstein et al., 2021), where arbitrary differentiable functions, parameterized by neural networks, are used to diffuse information on the graph, consequently learning a graph-level representation. This representation can then be used for various downstream tasks like node classification, link prediction, and graph classification. Different types of GNNs (Kipf & Welling, 2017; Hamilton et al., 2017; Veličković et al., 2018; Xu et al., 2019; Bodnar et al., 2021a,b; Bevilacqua et al., 2022), all tackling a variety of problems in various domains have been proposed with varied degree of success. Despite their widespread use, GNNs have a number of inherent problems. These include limited expressivity, (Leman, 1968; Morris et al., 2019), over-smoothing (Li et al., 2019; NT & Maehara, 2019; Oono & Suzuki, 2020; Zhou et al., 2021), and over-squashing (Alon & Yahav, 2021; Topping et al., 2022).

The phenomenon of over-squashing, first studied heuristically by Alon & Yahav (2021) and later theoretically formalized by Topping et al. (2022), is caused by the presence of structural bottlenecks in

---

*Equal contribution.

38th Conference on Neural Information Processing Systems (NeurIPS 2024).

the graph. These bottlenecks can be attributed to the first non-zero eigenvalue of the normalized graph Laplacian, also known as the spectral gap. The smaller the gap, the more susceptible a graph is to over-squashing. Recent work has explored rewiring the input graph to address these bottlenecks (Topping et al., 2022; Arnaiz-Rodríguez et al., 2022; Giraldo et al., 2023; Nguyen et al., 2023; Karhadkar et al., 2023), but suggest there has to be a trade-off between over-squashing and over-smoothing (Keriven, 2022). Instead, we propose to leverage the Braess paradox (Braess, 1968; Eldan et al., 2017) that posits certain edge *deletions* can maximize the spectral gap. We propose to approximate the spectral change in a computationally efficient manner by leveraging Matrix Perturbation Theory (Stewart & Sun, 1990). Our proposed framework allows us to jointly address the problem of over-squashing, by increasing the spectral gap, and over-smoothing, by *slowing* down the rate of smoothing. We find that our method is especially effective in heterophilic graph settings, where we delete edges between nodes of different labels, thus preventing unnecessary aggregation. We empirically show that our proposed method outperforms other graph rewiring methods on node classification and graph classification tasks. We also show that spectral gap based edge deletions can help identify graph lottery tickets (GLTs) (Frankle & Carbin, 2019), that is, sparse sub-networks that can match the performance of dense networks.

## 1.1 Contributions

1. Inspired by the Braess phenomenon, we prove that, contrary to common assumptions, over-smoothing and over-squashing are not necessarily diametrically opposed. By deriving a minimal example, we show that both can be mitigated by spectral based edge deletions.

2. Leveraging matrix perturbation theory, we propose a Greedy graph pruning algorithm (PROXYDELETE) that maximizes the spectral gap in a computationally efficient way. Similarly, our algorithm can also be utilized to add edges in a joint framework. We compare this approach with a novel graph rewiring scheme based on Eldan's criterion (Eldan et al., 2017) that provides guarantees for edge deletions and a stopping criterion for pruning, but is computationally less efficient.

3. Our results connect literature on three seemingly disconnected topics: over-smoothing, over-squashing, and *graph lottery tickets*, which explain observed improvements in generalization performance by graph pruning. Utilizing this insight, we demonstrate that graph sparsification based on our proxy spectral gap update can perform better than or on par with a contemporary baseline (Chen et al., 2021) that takes additional node features and labels into account. This highlights the feasibility of finding winning subgraphs at initialization.

## 2 Related work

**Over-squashing.** Alon & Yahav (2021); Topping et al. (2022) have observed that over-squashing, where information from distant nodes are not propagated due to topological bottlenecks in the graph, hampers the performance of GNNs. A promising line of work that attempts to alleviate this issue is *graph rewiring*. This task aims to modify the edge structure of the graph either by adding or deleting edges. Gasteiger et al. (2019) propose to add edges according to graph diffusion kernel, such as personalized PageRank, to rely less on messages from only one-hop neighbors, thus alleviating over-squashing. Topping et al. (2022) propose Stochastic Discrete Ricci Flow (SDRF) to rewire the graph based on curvature. Banerjee et al. (2022) resort to measuring the spectral expansion with respect to the number of rewired edges and propose a random edge flip algorithm that transforms the given input graph into an Expander graph. Contrarily, Deac et al. (2022) show that negatively curved edges might be inevitable for building scalable GNNs without bottlenecks and advocate the use of Expander graphs for message passing. Arnaiz-Rodríguez et al. (2022) introduces two new intermediate layers called CT-LAYER and GAP-LAYER, which can be interspersed between GNN layers. The layers perform edge re-weighting (which minimizes the gap) and introduce additional parameters. Karhadkar et al. (2023) propose FoSR, a graph rewiring algorithm that sequentially adds edges to maximize the first-order approximation of the spectral gap. A recent work by Black et al. (2023) explores the idea of characterizing over-squashing through the lens of effective resistance (Chandra et al., 1996). Giovanni et al. (2023) provide a comprehensive account of over-squashing and studies the interplay of depth, width and the topology of the graph.

**Over-smoothing.** It is a known fact that increasing network depth (He et al., 2016) often leads to better performance in the case of deep neural networks. However, naively stacking GNN layers often seems

to harm generalization. And one of the reasons is over-smoothing (Li et al., 2019; Oono & Suzuki, 2020; NT & Maehara, 2019; Zhou et al., 2021; Rusch et al., 2023a), where repeated aggregation leads to node features, in particular nodes with different labels, becoming indistinguishable. Current graph rewiring strategies, such as FoSR (Karhadkar et al., 2023), which rely on iteratively adding edges based on spectral expansion, may help mitigate over-squashing but also increase the smoothing induced by message passing. Curvature based methods such as Nguyen et al. (2023); Giraldo et al. (2023) aim to optimize the degree of smoothing by graph rewiring, as they assume that over-smoothing is the result of too much information propagation, while over-squashing is caused by too little. Within this framework, they assume that edge deletions always reduce the spectral gap. In contrast, we show and exploit that some deletions can also increase it. Furthermore, we rely on a different, well established concept of over-smoothing (Keriven, 2022) that also takes node features into account and is therefore not diametrically opposed to over-squashing. As we show, over-smoothing and over-squashing can be mitigated jointly. Moreover, we propose a computationally efficient approach to achieve this with spectral rewiring. In contrast to our proposal, curvature based methods (Nguyen et al., 2023; Giraldo et al., 2023) do not scale well to large graphs. For instance, Nguyen et al. (2023) propose a batch Ollivier-Ricci (BORF) curvature based rewiring approach to add and delete edges, which solves optimal transport problems and runs in cubic time.

**Graph sparsification and lottery tickets.** Most GNNs perform recursive aggregations of neighborhood information. This operation becomes computationally expensive when the graphs are large and dense. A possible solution for this is to extract a subset of the graph which is representative of the dense graph, either in terms of their node distribution (Eden et al., 2018) or graph spectrum (Adhikari et al., 2017). Zheng et al. (2020); Li et al. (2020) formulate graph sparsification as an optimization problem by resorting to learning surrogates and ADMM respectively. With the primary aim to reduce the computational resource requirements of GNNs, a line of work that transfers the lottery ticket hypothesis (LTH) by Frankle & Carbin (2019) to GNNs (Chen et al., 2021; Hui et al., 2023), prunes the model weights in addition to the adjacency matrix. The resulting *winning graph lottery ticket* (GLT) can match or surpass the performance of the original dense model. While our theoretical understanding of GLTs is primarily centered around their existence (Ferbach et al., 2022; Burkholz et al., 2022; Burkholz, 2022b,a), our insights inspired by the Braess paradox add a complementary lens to our understanding of how generalization can be improved, namely by reducing over-squashing and over-smoothing with graph pruning. So far, the spectral gap has only been employed to maintain a sufficient degree of connectivity of bipartite graphs that are associated with classic feed-forward neural network architectures (Pal et al., 2022; Hoang et al., 2023). We highlight that the spectral gap can also be employed as a pruning at initialization technique (Frankle et al., 2021) that does not take node features into account and can achieve computational resource savings while reducing the generalization error, which is in line with observations for random pruning of CNNs (Gadhikar et al., 2023; Gadhikar & Burkholz, 2024).

## 3 Theoretical insights into spectral rewiring

To prove our claim that over-smoothing and over-squashing can both be alleviated jointly, we provide a minimal example as illustrated in Figure 1. Utilizing the Braess paradox, we achieve this by the deletion of an edge. In contrast, an edge addition that addresses over-squashing still causes over-smoothing, yet less drastically than another edge addition that worsens over-squashing.

**Reducing over-squashing via the spectral gap.** From a spectral perspective, bottlenecks, which hamper the information flow by over-squashing, can be characterized by the spectral gap of the (symmetric) normalized graph Laplacian $\mathcal{L}_\mathcal{G}$, where $\mathcal{G} = (\mathcal{V}, \mathcal{E})$. The Laplacian of the graph is $\mathcal{L} = D - A$, where $A$ is the adjacency matrix and $D$ the diagonal degree matrix. The symmetric normalized graph Laplacian is defined as $\mathcal{L}_\mathcal{G} = D^{-1/2}\mathcal{L}D^{-1/2}$. Let $\{\lambda_0 < \lambda_1 < \lambda_2, ...\lambda_n\}$ be the eigenvalues of $\mathcal{L}_\mathcal{G}$ arranged in ascending order and let $\lambda_1(\mathcal{L}_\mathcal{G})$ be the first non-zero eigenvalue of the normalized graph Laplacian, which is also called the spectral gap of the graph. For a graph where distant network components are connected only by a few bridging edges, all the information has to be propagated via these edges. The information flow through edges is encoded by the Cheeger (1971) constant $h_S = \min_{S \subset V} \frac{|\partial S|}{\min\{Vol(S), Vol(S \setminus V)\}}$ where $\partial S = \{(u, v) : u \in S, v \in \mathcal{V} \setminus S\}$ and $Vol(S) = \sum_{u \in S} d_u$, being $d_u$ the degree of the node $u$. The spectral gap is bounded by the Cheeger inequality $2h_\mathcal{G} \geq \lambda_1 \geq \frac{h_\mathcal{G}^2}{2}$, which motivates it as a measure of over-squashing.

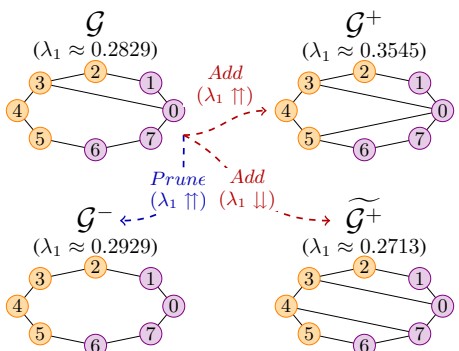

Figure 1: Braess' paradox. We derive a simple example where deleting an edge from $\mathcal{G}$ to obtain $\mathcal{G}^-$ yields a higher spectral gap. Alternatively, we add a single edge to the base graph to either increase ($\mathcal{G}^+$) or to decrease ($\widetilde{\mathcal{G}^+}$) the spectral gap. The relationship between the four graphs is highlighted by arrows when an edge is added/deleted.

**Braess' paradox.** Braess (1968) found a counter-intuitive result for road networks: even if all travelers behave selfishly, the removal of a road can still improve each of their individual travel times. That is, there is a violation of monotonicity in the traffic flow with respect to the number of edges of a network. For instance, Chung & Young (2010) has shown that Braess' paradox occurs with high probability in Erdős-Rényi random graphs, and Chung et al. (2012) have confirmed it for a large class of Expander graphs. The paradox can be analogously applied to related graph properties such as the spectral gap of the normalized Laplacian. Eldan et al. (2017) have studied how the spectral gap of a random graph changes after edge additions or deletions, proving a strictly positive occurrence of the paradox for typical instances of ER graphs. This result inspires us to develop an algorithm for rewiring a graph by specifically eliminating edges that increase this quantity, which we can expect to carry out with high confidence in real-world graphs. Their Lemma 3.2 (when reversed) states a sufficient condition that guarantees a spectral gap increase in response to a deletion of an edge.

**Lemma 3.1.** *Eldan et al. (2017): Let $\mathcal{G} = (\mathcal{V}, \mathcal{E})$ be a finite graph, with $f$ denoting the eigenvector and $\lambda_1(\mathcal{L}_\mathcal{G})$ the eigenvalue corresponding to the spectral gap. Let $\{u,v\} \notin \mathcal{V}$ be two vertices that are not connected by an edge. Denote $\hat{\mathcal{G}} = (\mathcal{V}, \hat{\mathcal{E}})$, the new graph obtained after adding an edge between $\{u,v\}$, i.e., $\hat{\mathcal{E}} := \mathcal{E} \cup \{u,v\}$. Denote with $\mathcal{P}_f := \langle f, \hat{f}_0 \rangle$ the projection of $f$ onto the top eigenvector of $\hat{\mathcal{G}}$. Define $g(u, v, \mathcal{L}_\mathcal{G}) :=$*

$$-\mathcal{P}_f^2 \lambda_1(\mathcal{L}_\mathcal{G}) - 2(1 - \lambda_1(\mathcal{L}_\mathcal{G})) \left( \frac{\sqrt{d_u + 1} - \sqrt{d_u}}{\sqrt{d_u + 1}} f_u^2 + \frac{\sqrt{d_v + 1} - \sqrt{d_v}}{\sqrt{d_v + 1}} f_v^2 \right) + \frac{2 f_u f_v}{\sqrt{d_u + 1}\sqrt{d_v + 1}}.$$

*If $g(u, v, \mathcal{L}_\mathcal{G}) > 0$, then $\lambda_1(\mathcal{L}_\mathcal{G}) > \lambda_1(\mathcal{L}_{\hat{\mathcal{G}}})$.*

As a showcase example of the Braess phenomenon, let us analyze the behaviour of the spectral gap in terms of an edge perturbation on the ring graph of $n$ nodes $R_n$. We consider the ring $R_8$ as $\mathcal{G}^-$, the deletion of an edge from graph $\mathcal{G}$ in Figure 1.

**Proposition 3.2.** *The spectral gap of $\mathcal{G}$ increases with the deletion of edge $\{0,3\}$, i.e., $\lambda_1(\mathcal{L}_{\mathcal{G}^-}) > \lambda_1(\mathcal{L}_\mathcal{G})$. It also increases with the addition of edge $\{0,5\}$ or decreases with the addition of edge $\{4,7\}$, i.e., $\lambda_1(\mathcal{L}_{\mathcal{G}^+}) > \lambda_1(\mathcal{L}_\mathcal{G})$ and $\lambda_1(\mathcal{L}_{\widetilde{\mathcal{G}^+}}) < \lambda_1(\mathcal{L}_\mathcal{G})$.*

We leverage Eldan's Lemma 3.1 in Appendix A.1 and apply the spectral graph proxies in our derivations starting from an explicit spectral analysis of the ring graph. While these derivations demonstrate that we can reduce over-squashing (i.e., increase the spectral gap) by edge deletions, we show next that edge deletions can also alleviate over-smoothing.

**Slowing detrimental over-smoothing.** For GNNs with mean aggregation, increasing the spectral gap usually promotes smoothing and thus leads to higher node feature similarity. Equating a high node feature similarity with over-smoothing would thus imply a trade-off between over-smoothing and over-squashing. Methods by Giraldo et al. (2023); Nguyen et al. (2023) seek to find the right amount of smoothing by adding edges to increase the gap and deleting edges to decrease it. *Contrarily, we*

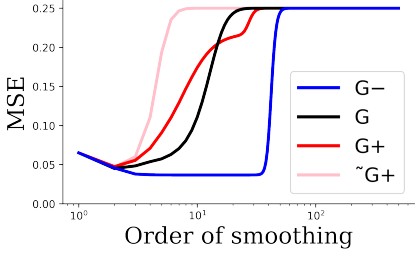
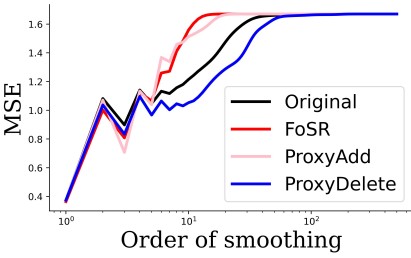

(a) Smoothing test for graphs in Figure 1.   (b) Smoothing test for the Texas dataset.

Figure 2: We plot the MSE vs order of smoothing for our four synthetic graphs (2(a)), and for a real heterophilic dataset with the result of different rewiring algorithms to it: FoSR (Karhadkar et al., 2023) and PROXYADD for adding (200 edges), and our PROXYDELETE for deleting edges (5 edges) (2(b)). We find that deleting edges helps reduce over-smoothing, while still mitigating over-squashing via the spectral gap increase.

*argue that deleting edges can also increase the gap while adding edges could decrease it*, as our previous analysis demonstrates. Thus, both edge deletions and additions allow to control which node features are aggregated, while mitigating over-squashing. Such node features are central to a more nuanced concept of over-smoothing that acknowledges that increasing the similarity of nodes that share the same label, while keeping nodes with different labels distinguishable, aids the learning task.

To measure over-smoothing, we adopt the Linear GNN test bed proposed by Keriven (2022), which uses a linear ridge regression (LRR) setup with mean squared error (MSE) as the loss. We assign two classes to nodes according to their color in Figure 1, and one-dimensional features that are drawn independently from normal distributions $\mathcal{N}(1, 1)$ and $\mathcal{N}(-1, 1)$, respectively. Figure 2(a) compares how our exemplary graphs (see Figure 1) influence over-smoothing in this setting. While adding edges can accelerate the rate of smoothing, pruning strikingly aids in reducing over-smoothing —and still reduces over-squashing by increasing the spectral gap. Note that the real world heterophilic graph example shows a similar trend and highlights the utility of the spectral pruning algorithm PROXYDELETE, which we describe in the next section, over edge additions by the strong baseline FoSR. Additional real world examples along with cosine distance between nodes of different labels before and after spectral pruning and plots for Dirichlet energy can be found in Appendix D.

In the following, we discuss and analyze rigorously the reasons for this finding. Consider again the ring graph $\mathcal{G}^-$, which has an *inter-class* edge pruned from our base graph $\mathcal{G}$; this avoids a problematic aggregation step and in this way mitigates over-smoothing. Instead of deleting an edge, we could also add an edge arriving at $\mathcal{G}^+$, which would lead to a higher spectral gap than the edge deletion. Yet, it adds an edge between nodes with different labels and therefore leads to over-smoothing. We also prove this relationship rigorously for one step of mean aggregation.

**Proposition 3.3.** *As more edges are added (from $\mathcal{G}^-$ to $\mathcal{G}$, or from $\mathcal{G}$ to $\mathcal{G}^+$ or $\widetilde{\mathcal{G}^+}$), the average value over same-class node representations after a mean aggregation round becomes less informative.*

The proof is presented in Appendix A.2. We argue that similar situations arise particularly in heterophilic learning tasks, where spectral gap optimization would frequently delete inter-class edges but also add inter-class edges. Thus, mostly edge deletions can mitigate over-squashing and over-smoothing simultaneously.

Clearly, this argument relies on the specific distribution of labels. Other scenarios are analyzed in Appendix B to also highlight potential limitations of spectral rewiring that does not take node labels into account.

Following this argument, however, we could ask if the learning task only depends on the label distribution. The following proposition highlights why spectral gap optimization is justified beyond label distribution considerations.

**Proposition 3.4.** *After one round of mean aggregation, the node features of $\mathcal{G}^+$ are more informative compared to $\widetilde{\mathcal{G}^+}$.*

Note that $\widetilde{\mathcal{G}^+}$ decreases the spectral gap, while $\mathcal{G}^+$ increases it relative to $\mathcal{G}$. However, the label configuration of $\widetilde{\mathcal{G}^+}$ seems more advantageous because, for the changed nodes, the number of neighbors of the same class label remains in the majority in contrast to $\mathcal{G}^+$. Still, the spectral gap increase seems to aid the learning task compared to the spectral gap decrease.

## 4   Braess-inspired graph rewiring

We introduce two algorithmic approaches to perform spectral rewiring. Our main proposal is computationally more efficient and more effective in spectral gap approximation than baselines, as we also showcase in Table 14. The other approach based on Eldan's Lemma is also analyzed, as it provides theoretical guarantees for edge deletions. However, it does not scale well to larger graphs.

**Greedy approach to modify edges.** Evaluating all potential subsets of edges that we could add or delete is computationally infeasible due to the combinatorially exploding number of possible candidates. Therefore, we resort to a Greedy approach, in which we add or delete a single edge iteratively. In every iteration, we rank candidate edges according to a proxy of the spectral gap change that would be induced by the considered rewiring operation, as described next.

### 4.1   Graph rewiring with Proxy spectral gap updates

**Update of eigenvalues and eigenvectors.** Calculating the eigenvalues for every normalized graph Laplacian obtained by the inclusion or exclusion of a single edge would be a highly costly method. The ability to use the spectral gap directly as a criterion to rank edges requires a formula to efficiently estimate it for one edge flip. For this we resort to Matrix Perturbation Theory (Stewart & Sun, 1990; von Luxburg, 2007) to capture the change in eigenvalues and eigenvectors approximately. Our update scheme is similar to the proposal by Bojchevski & Günnemann (2019) in the context of adversarial flips. The change in the eigenvalue and eigenvector for a single edge flip $(u, v)$ is given by

$$\acute{\lambda} \approx \lambda + \Delta w_{u,v}((f_u - f_v)^2 - \lambda(f_u^2 + f_v^2)), \tag{1}$$

where $\lambda$ is the initial eigenvalue; $\{f_u, f_v\}$ are entries of the leading eigenvector, $\Delta w_{u,v} = 1$ if we add an edge and $\Delta w_{u,v} = -1$ if we delete an edge. Note that this proxy is only used to rank edges efficiently. After adding/deleting the top $M$ edges (where $M = 1$ in our experiments), we update the eigenvector and the spectral gap by performing a few steps of power iteration. To this end, we initialize the function `eigsh` of the scipy sparse library in Python, which is based on the Implicitly Restarted Lanczos Method (Lehoucq et al., 1998), with our current estimate of the leading eigenvector. Both our resulting algorithms, PROXYDELETE for deleting edges and PROXYADD for adding edges, are detailed in Appendix C.

**Time Complexity of PROXYDELETE.** The algorithm runs in $\mathcal{O}(N \cdot (|\mathcal{E}| + s(\mathcal{G})))$ where $N$ is the number of edges to delete, and $s(\mathcal{G})$ denotes the complexity of the algorithm that updates the leading eigenvector and eigenvalue at the end of every iteration. In our setting, this requires a constant number of power method iterations, which is of complexity $s(\mathcal{G}) = O(|\mathcal{E}|)$. Note that, because we choose to only delete one edge, the ranking does not need to be sorted to obtain its maximum. By having an $\mathcal{O}(1)$ proxy measure to score candidate edges, we are able to improve the overall runtime complexity from the original $\mathcal{O}(N \cdot |\mathcal{E}| \cdot s(\mathcal{G}))$. Furthermore, even though this does not impact the asymptotic complexity, deleting edges instead of adding them makes every iteration run on a gradually smaller graph, which can further induce computational savings for the downstream task.

**Time Complexity of PROXYADD.** The run time analysis consists of the same elements as the edge deletion algorithm. The key distinction is that the ranking is conducted on the complement of the graph's edges, $\bar{\mathcal{E}}$. Since the set of missing edges is usually larger than the existing edges in real world settings, to save computational overhead, it is possible to only sample a constant amount of edges. See Section F for empirical runtimes.

### 4.2   Graph rewiring with Eldan's criterion

Lemma 3.1 states a sufficient condition for the Braess paradox. It naturally defines a scoring function of edges to rank them according to their potential to maximize the spectral gap based on the function $g$. However, the computation of this ranking is significantly more expensive than other considered

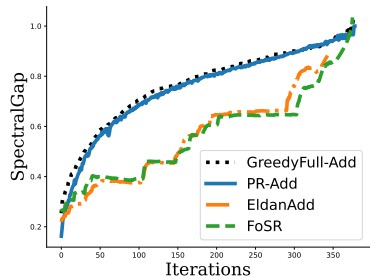
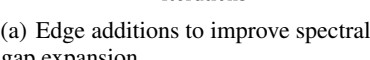
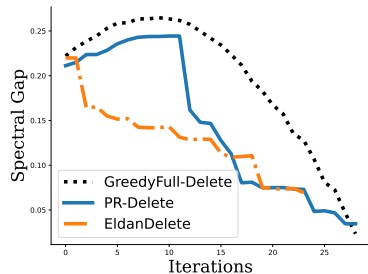

(a) Edge additions to improve spectral gap expansion.

(b) Edge deletions to improve spectral gap expansion.

Figure 3: We instantiate a toy ER graph with 30 nodes and 58 edges. We compare FoSR (Karhadkar et al., 2023), our proxy spectral gap based methods, and our Eldan's criterion based edge methods.

Table 1: Results on Long Range Graph Benchmark datasets.

| Method | PascalVOC-SP (Test F1 ↑) | Peptides-Func (Test AP ↑) | Peptides-Struct (Test MAE ↓) |
|---|---|---|---|
| Baseline-GCN | 0.1268±0.0060 | 0.5930±0.0023 | 0.3496±0.0013 |
| DRew+GCN | 0.1848±0.0107 | **0.6996±0.0076** | 0.2781±0.0028 |
| FoSR+GCN | 0.2157±0.0057 | 0.6526±0.0014 | 0.2499±0.0006 |
| ProxyAdd+GCN | **0.2213±0.0011** | 0.6789±0.0002 | **0.2465±0.0004** |
| ProxyDelete+GCN | 0.2170±0.0015 | 0.6908±0.0007 | **0.2470±0.0080** |

algorithms, as each scoring operation needs access to the leading eigenvector of the perturbed graph with an added or deleted edge. In case of edge deletions, we also need to approximate the spectral gap similar to our Proxy algorithms. As the involved projection $\mathcal{P}_f$ is a dot product of eigenvectors, it requires $\mathcal{O}(|\mathcal{V}|)$ operations. Even though this algorithm does not scale well to large graphs without focusing on a small random subset of candidate edges, we still consider it as baseline, as it defines a more conservative criterion to assess when we should stop deleting edges. The precise algorithms are stated in Appendix C.

### 4.3 Approximation quality

To check whether the proposed edge modification algorithms are indeed effective in the spectral gap expansion, we conduct experiments on an Erdös-Rényi (ER) graph with $(|\mathcal{V}|, |\mathcal{E}|) = (30, 58)$ in Figure 3. Our ideal baseline that scores each candidate with the correct spectral gap change would usually be computationally too expensive, because each edge scoring requires $O(|\mathcal{E}|)$ computations. For our small synthetic test bed, we still compute it to assess the approximation quality of the proposed algorithms, and of the competitive baseline FoSR (Karhadkar et al., 2023). For both edge additions (Figure 3(a)) and deletions (Figure 3(b)), we observe that the Proxy method outlined in Algorithm 1 usually leads to a better spectral expansion approximation. In addition, we report the spectral gaps that different methods obtain on real world data in Table 16 in the Appendix, which highlights that our proposals are consistently most effective in increasing the spectral gap.

## 5 Experiments

### 5.1 Long Range Graph Benchmark

The Long Range Graph Benchmark (LRGB) was introduced by Dwivedi et al. (2023) specifically to create a test bed for over-squashing. We compare our proposed PROXYADD and PROXYDELETE methods with DRew (Gutteridge et al., 2023), a recently proposed strong baseline for addressing over-squashing using a GCN as our backbone architecture in Table 1. We adopt the experimental setting of Tönshoff et al. (2023), we adopt DRew baseline results from the original paper. We evaluate on the following datasets and tasks: 1) PascalVOC-SP - Semantic image segmentation as a node classification task operating on superpixel graphs. 2) Peptides-func - Peptides modeled as molecular

Table 2: Node classification on Roman-Empire dataset.

| Method | #EdgesAdded | Accuracy | #EdgesDeleted | Accuracy | Layers |
|---|---|---|---|---|---|
| GCN | - | 70.30±0.73 | - | 70.30±0.73 | 5 |
| GCN+FoSR | 50 | 73.60±1.11 | - | - | 5 |
| GCN+Eldan | 50 | 72.11±0.80 | 50 | **79.14±0.73** | 5 |
| GCN+ProxyGap | 50 | **77.54±0.74** | 20 | 77.45±0.68 | 5 |
| GAT | - | 80.89±0.70 | - | 80.89±0.70 | 5 |
| GAT+FoSR | 50 | 81.88±1.07 | - | - | 5 |
| GAT+Eldan | 50 | 81.13±0.50 | 100 | 82.12±0.69 | 5 |
| GAT+ProxyGap | 50 | **86.07±0.46** | 20 | **86.00±0.36** | 5 |
| GCN | - | 68.89±0.77 | - | 68.89±0.77 | 10 |
| GCN+FoSR | 100 | 73.85±1.26 | - | - | 10 |
| GCN+Eldan | 100 | 75.39±0.96 | 100 | **80.40±0.54** | 10 |
| GCN+ProxyGap | 20 | 78.31±0.47 | 20 | 78.19±0.71 | 10 |
| GAT | - | 80.23±0.59 | - | 80.23±0.59 | 10 |
| GAT+FoSR | 100 | 81.37±1.14 | - | - | 10 |
| GAT+Eldan | 100 | **87.19±0.38** | 20 | 86.90±0.37 | 10 |
| GAT+ProxyGap | 20 | 83.45±0.42 | 20 | 86.44±0.40 | 10 |
| GCN | - | 67.77±0.90 | - | 67.77±0.90 | 20 |
| GCN+FoSR | 100 | 75.14±1.43 | - | - | 20 |
| GCN+Eldan | 100 | 75.52±1.16 | 20 | **80.37±0.70** | 20 |
| GCN+ProxyGap | 50 | 77.96±0.65 | 20 | 78.03±0.71 | 20 |
| GAT | - | 79.22±0.70 | - | 79.22±0.70 | 20 |
| GAT+FoSR | 100 | 80.64±1.12 | - | 80.64±1.12 | 20 |
| GAT+Eldan | 100 | **86.79±0.58** | 50 | 86.70±0.50 | 20 |
| GAT+ProxyGap | 10 | 86.25±0.63 | 20 | 86.15±0.61 | 20 |

Table 3: Node classification on Amazon-Ratings.

| Method | #EdgesAdded | Accuracy | #EdgesDeleted | Accuracy | Layers |
|---|---|---|---|---|---|
| GCN | - | 47.20±0.33 | - | 47.20±0.33 | 10 |
| GCN+FoSR | 25 | 49.68±0.73 | - | - | 10 |
| GCN+Eldan | 25 | 48.71±0.99 | 100 | **50.15±0.50** | 10 |
| GCN+ProxyGap | 10 | 49.72±0.41 | 50 | **49.75±0.46** | 10 |
| GAT | - | 47.43±0.44 | - | 47.43±0.44 | 10 |
| GAT+FoSR | 25 | 51.36±0.62 | - | - | 10 |
| GAT+Eldan | 25 | 51.68±0.60 | 50 | **51.80±0.27** | 10 |
| GAT+ProxyGap | 20 | 49.06±0.92 | 100 | **51.72±0.30** | 10 |
| GCN | - | 47.32±0.59 | - | 47.32±0.59 | 20 |
| GCN+FoSR | 100 | 49.57±0.39 | - | - | 20 |
| GCN+Eldan | 50 | **49.66±0.31** | 20 | 48.32±0.76 | 20 |
| GCN+ProxyGap | 50 | 49.48±0.59 | 500 | **49.58±0.59** | 20 |
| GAT | - | 47.31±0.46 | - | 47.31±0.46 | 20 |
| GAT+FoSR | 100 | 51.31±0.44 | - | - | 20 |
| GAT+Eldan | 20 | 51.40±0.36 | 20 | **51.64±0.44** | 20 |
| GAT+ProxyGap | 50 | 47.53±0.90 | 20 | **51.69±0.46** | 20 |

Table 4: Node classification on Minesweeper.

| Method | #EdgesAdded | Accuracy | #EdgesDeleted | Test ROC | Layers |
|---|---|---|---|---|---|
| GCN | - | 88.57± 0.64 | - | 88.57± 0.64 | 10 |
| GCN+FoSR | 50 | 90.15±0.55 | - | - | 10 |
| GCN+Eldan | 100 | **90.11±0.50** | 50 | 89.49±0.60 | 10 |
| GCN+ProxyGap | 20 | 89.59±0.50 | 20 | 89.57±0.49 | 10 |
| GAT | - | 93.60±0.64 | - | 93.60±0.64 | 10 |
| GAT+FoSR | 100 | 93.14±0.43 | - | - | 10 |
| GAT+Eldan | 50 | 93.26±0.48 | 100 | **93.82±0.56** | 10 |
| GAT+ProxyGap | 20 | 93.60±0.69 | 20 | **93.65±0.84** | 10 |
| GCN | - | 87.41±0.65 | - | 87.41±0.65 | 20 |
| GCN+FoSR | 100 | 89.64±0.55 | - | - | 20 |
| GCN+Eldan | 72 | **89.70±0.57** | 10 | 88.90±0.44 | 20 |
| GCN+ProxyGap | 20 | 89.46±0.50 | 50 | 89.35±0.30 | 20 |
| GAT | - | 93.92±0.52 | - | 93.92±0.52 | 20 |
| GAT+FoSR | 50 | 93.56±0.64 | - | - | 20 |
| GAT+Eldan | 10 | 93.92±0.44 | 20 | **95.48±0.64** | 20 |
| GAT+ProxyGap | 20 | **94.89±0.67** | 20 | 94.64±0.81 | 20 |

graphs. The task is graph classification. 3) Peptides-struct - Peptides modeled as molecular graphs. The task is to predict various molecular properties, hence a graph regression task.

The top performance is highlighted in bold. Evidently, our proposed rewiring methods outperform DRew (Gutteridge et al., 2023) and FoSR (Karhadkar et al., 2023) on PascalVOC and Peptides-struct, and achieves comparable performance on Peptides-func.

In addition, Table 10 in the appendix compares different rewiring strategies for node classification on other commonly used datasets and graph classification (§E.2) for adding edges, since FoSR (Karhadkar et al., 2023) was primarily tested on this task.

**Node classification on large heterophilic datasets.** Platonov et al. (2023) point out that most progress on heterophilic datasets is unreliable since many of the used datasets have drawbacks, including duplicate nodes in Chameleon and Squirrel datasets, which lead to train-test data leakage. The sizes of the small graph sizes also lead to high variance in the obtained accuracies. Consequently, we also test our proposed algorithms on 3/5 of their newly introduced larger datasets and use GCN (Kipf & Welling, 2017) and GAT (Veličković et al., 2018) as our backbone architectures. As a higher depth potentially increases over-smoothing, we also analyze how our methods fares with varied number of layers. To that end, we adopt the code base and experimental setup of Platonov et al. (2023); the datasets are divided into 50/25/25 split for train/test/validation respectively. The test accuracy is reported as an average over 10 runs. To facilitate training deeper models, skip connections and layer normalization are employed. We compare FoSR (Karhadkar et al., 2023) and our proposals based on the Eldan criterion as well as PROXYADD and PROXYDELETE in Tables 2,3,4. The top performance is highlighted in **bold**. Evidently, for increasing depth, even though the GNN performance should degrade because of over-smoothing, we achieve a significant boost in accuracy compared to baselines, which we attribute to the fact that our methods delete inter-class edges —thus slowing down detrimental smoothing.

Table 5: Pruning for lottery tickets comparing UGS to our ELDANDELETE pruning and our PROXYDELETE pruning. We report Graph Sparsity (GS), Weight Sparsity (WS), and Accuracy (Acc).

| Method | Cora | | | Citeseer | | | Pubmed | | |
|---|---|---|---|---|---|---|---|---|---|
| Metrics | GS | WS | Acc | GS | WS | Acc | GS | WS | Acc |
| UGS | 79.85% | 97.86% | 68.46±1.89 | 78.10% | 97.50% | **66.50±0.60** | 68.67% | 94.52% | 76.90±1.83 |
| ELDANDELETE-UGS | 79.70% | 97.31% | 68.73±0.01 | 77.84% | 96.78% | 64.60±0.00 | 70.11% | 93.17% | **78.00±0.42** |
| PROXYDELETE-UGS | 78.81% | 97.24% | **69.26±0.63** | 77.50% | 95.83% | 65.43±0.60 | 78.81% | 97.24% | 75.25±0.25 |

**Pruning for graph lottery tickets.** In Sections §3 and §5, we have shown that graph pruning can improve generalization, mitigate over-squashing and also help slow down the rate of smoothing. Can we also use our insights to find lottery tickets (Frankle & Carbin, 2019)?

**To what degree is graph pruning feature data dependent?** The first extension of the Lottery Ticket Hypothesis to GNNs, called Unified Graph Sparsification (UGS) (Chen et al., 2021), prunes connections in the adjacency matrix and model weights that are deemed less important for a prediction task. Note that UGS relies on information that is obtained in computationally intensive prune-train cycles that take into account the data and the associated masks. In the context of GNNs, the input graph plays a central role in determining a model's performance at a downstream task. Naively pruning the adjacency matrix without characterizing what constitutes *important edges* is a pitfall we would want to avoid (Hui et al., 2023), yet resorting to expensive train-prune-rewind cycles to identify importance is also undesirable. This brings forth the questions: To what extent does the pruning criterion need to depend on the data? Is it possible to formulate a data/feature agnostic pruning criterion that optimizes a more general underlying principle to find lottery tickets? Morcos et al. (2019) and Chen et al. (2020) show, in the context of computer vision and natural language processing respectively, that lottery tickets can have universal properties that can even provably (Burkholz et al., 2022) transfer to related tasks.

**Lottery tickets that rely on the spectral gap.** However, even specialized structures need to maintain and promote information flow through their connections. This fact has inspired works like Pal et al. (2022); Hoang et al. (2023) to analyze how well lottery ticket pruning algorithms maintain the Ramanujan graph property of bipartite graphs, which is intrinsically related to the Cheeger constant and thus the spectral gap. They have further shown that rejecting pruning steps that would destroy a proxy of this property can positively impact the training process.

In the context of GNNs, we show that we can base the graph pruning decision even entirely on the spectral gap, but rely on a computationally cheaper approach to obtain a proxy. By replacing the magnitude pruning criterion for the graph with the Eldan criterion and PROXYDELETE to prune edges, in principle, we can avoid the need for additional data features and labels. This has the advantage that we could also prune the graph at initialization and thus benefit from the computational savings from the start. We use our proposed methods to prune the graph at initialization to the requisite sparsity level and then feed it to the GNN where the weights are pruned in an iterative manner. Our results are presented in Table 18, where we compare IMP based UGS (Chen et al., 2021) with our methods for different graph and weight sparsity levels. Note that, even though our method does not take any feature information into account and prunes purely based on the graph structure, our results are comparable. For datasets like Pubmed, we even slightly outperform the baseline. Table 5 shows results for jointly pruning the graph and parameter weights, which leads to better results due to potential positive effects of overparameterization on training (Gadhikar & Burkholz, 2024).

**Stopping criterion.** The advantage of using spectral gap based pruning (especially the Eldan criterion) is patent: It helps identify problematic edges that cause information bottlenecks and provides a framework to prune those edges. Unlike UGS, our proposed framework also has the advantage that we can couple the overall pruning scheme with a stopping criterion that follows naturally from our setup. We stop pruning the input graph when no available edges satisfy our criterion anymore.

# 6 Conclusion

Our work connects two seemingly distinct branches of the literature on GNNs: rewiring graphs to mitigate over-squashing and pruning graphs for lottery tickets to save computational resources.

Contributing to the first branch, we highlight that, contrary to the standard rewiring practice, not only adding but also pruning edges can increase the spectral gap of a graph exploiting the Braess paradox. By providing a minimal example, we prove that this way it is possible to address over-squashing and over-smoothing simultaneously. Experiments on large-scale heterophilic graphs confirm the practical utility of this insight. Contributing to the second branch, these results explain how pruning graphs moderately can improve the generalization performance of GNNs, in particular for heterophilic learning tasks. To utilize these insights, we have proposed a computationally efficient graph rewiring framework, which also induces a competitive approach to prune graphs for lottery tickets at initialization.

## Acknowledgments and Disclosure of Funding

We gratefully acknowledge funding from the European Research Council (ERC) under the Horizon Europe Framework Programme (HORIZON) for proposal number 101116395 SPARSE-ML.

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

# A  Proofs

## A.1  Proof of Proposition 3.2

**Spectral analysis for general $n$.**

For all $n$, the (normalized) Laplacian matrix of $R_n$ is circulant: all rows consist of the same elements, and each row is shifted to the right with respect to the previous one. The first row of $\mathcal{L}(R_n)$ is $r_n = \left(1, -\frac{1}{2}, 0, \ldots, 0, -\frac{1}{2}\right)$. All circulant matrices satisfy that their eigenvectors are made up of powers of the $n$th-roots of unity, and that its eigenvalues are the DFT of the matrix's first row (Gray, 2005). With this we easily obtain that its spectral gap is

$$\lambda_1 = \sum_{k=0}^{n-1} r_n(k) \cdot e^{-i2\pi\frac{k}{n}} = 1 - \frac{1}{2}\left(e^{-i2\pi\frac{1}{n}} + e^{-i2\pi\frac{-1}{n}}\right) = 1 - \cos\left(\frac{2\pi}{n}\right).$$

As stated before, one possible set of eigenvectors is $\omega_j(k) = \exp\left(i\frac{2\pi jk}{n}\right)$. Because their conjugates and their linear combinations are also eigenvectors, we can get real eigenvectors as $x_j(k) = \frac{\omega_j(k) - \omega_{-j}(k)}{2i} = \sin\frac{2\pi jk}{n}$. Alternatively, we can get $y_j(k) = \frac{\omega_j(k) + \omega_{-j}(k)}{2} = \cos\frac{2\pi jk}{n}$.

We only need to focus on the (pair of) eigenvectors for $j = 1$. Note that they are orthogonal to each other. Because they are both eigenvectors with the same eigenvalue $\lambda_1$, all linear combinations of them will also be eigenvectors with eigenvalue $\lambda_1$. This multiplicity lets us choose any of these vectors to fulfill Eldan's criterion. A limitation of our algorithm is that, in cases of multiplicity, we can only choose one of them, potentially giving that edge a disadvantage —the Lemma holds as long as there exists one that fulfills it, but not necessarily the one we have chosen.

The norms of $x_1$ and $y_1$ are $\sqrt{\frac{n}{2}}$. Therefore, the norm of any linear combination of them is $\|\mu x_1 + \nu y_1\| = \sqrt{\frac{n}{2}}\sqrt{\mu^2 + \nu^2}$. We denote the normalized linear combination of $x_1$ and $y_1$ as

$$f_1^{(\mu,\nu)} = \frac{\sqrt{2}(\mu x_1 + \nu y_1)}{\sqrt{n(\mu^2 + \nu^2)}}$$

Our choice will be $\mu = 3$, $\nu = 1$, i.e., $f_1^{(3,1)} = \frac{(3x_1 + y_1)}{\sqrt{5n}}$.

**Elements of the criterion for general $n$.** As per Eldan et al. (2017), the first eigenvector of the new graph's normalized Laplacian is $\hat{f}_0 = \hat{D}^{\frac{1}{2}} \mathbb{1}/\sqrt{\sum \hat{d}_i}$. In our case: $\hat{f}_0(k) = \frac{\sqrt{3}}{\sqrt{2(n+1)}}$ if $k \in \{u, v\}$, and $\frac{\sqrt{2}}{\sqrt{2(n+1)}}$ if $k \notin \{u, v\}$. With it we calculate the projection, dependent on the eigenvector $f_1$:

$$\mathcal{P}_{f_1} = \sum_{k=0}^{n-1} f_1(k)\hat{f}_0(k) = \sum_{k=0,\ k\neq u,v}^{n-1} \frac{\sqrt{2}}{\sqrt{2(n+1)}} f_1(k) + \frac{\sqrt{3}}{\sqrt{2(n+1)}}\left(f_1(u) + f_1(v)\right)$$

$$= \frac{\sqrt{3} - \sqrt{2}}{\sqrt{2(n+1)}}\left(f_1(u) + f_1(v)\right).$$

We also have, for all $n$, $u$ and $v$, that $\frac{\sqrt{d_u+1}-\sqrt{d_u}}{\sqrt{d_u+1}} = \frac{\sqrt{d_v+1}-\sqrt{d_v}}{\sqrt{d_v+1}} = \frac{\sqrt{3}-\sqrt{2}}{\sqrt{3}} = 1 - \sqrt{\frac{2}{3}}$. We can update the criterion with these considerations: $g(u, v, R_n) =$

$$= -\mathcal{P}_{f_1}^2 \lambda_1 - 2(1 - \lambda_1)\left(\frac{\sqrt{d_u+1}-\sqrt{d_u}}{\sqrt{d_u+1}} f_1(u)^2 + \frac{\sqrt{d_v+1}-\sqrt{d_v}}{\sqrt{d_v+1}} f_1(v)^2\right) + \frac{2f_1(u)f_1(v)}{\sqrt{d_u+1}\sqrt{d_v+1}}$$

$$= -\left(\frac{\sqrt{3}-\sqrt{2}}{\sqrt{2(n+1)}}\right)^2\left(1 - \cos\left(\frac{2\pi}{n}\right)\right)(f_1(u) + f_1(v))^2$$

$$- 2\cos\left(\frac{2\pi}{n}\right)\left(1 - \sqrt{\frac{2}{3}}\right)\left(f_1(u)^2 + f_1(v)^2\right) + \frac{2f_1(u)f_1(v)}{3}.$$

**Case $n = 8$, $(u, v) = \{0, 3\}$.** We choose $f_1 := f_1^{(3,1)} = \frac{(3x_1 + y_1)}{\sqrt{40}}$. We have $f_1(0) = \frac{1}{2\sqrt{10}}$ and $f_1(3) = \frac{1}{2\sqrt{5}}$. Then:

$$(f_1(u) + f_1(v))^2 = \left(\frac{1}{2\sqrt{10}} + \frac{1}{2\sqrt{5}}\right)^2 = \frac{3 + 2\sqrt{2}}{40}$$

$$(f_1(u)^2 + f_1(v)^2) = \left(\frac{1}{2\sqrt{10}}\right)^2 + \left(\frac{1}{2\sqrt{5}}\right)^2 = \frac{3}{40}$$

$$f_1(u)f_1(v) = \frac{1}{2\sqrt{10}}\frac{1}{2\sqrt{5}} = \frac{\sqrt{2}}{40}$$

Finally, Eldan's criterion for $n = 8$, $(u, v) = \{0, 3\}$, and our choice of $f_1$ is $g(0, 3, R_8) =$

$$= -\left(\frac{\sqrt{3} - \sqrt{2}}{\sqrt{18}}\right)^2 \left(1 - \cos\left(\frac{\pi}{4}\right)\right)(f_1(u) + f_1(v))^2 - 2\cos\left(\frac{\pi}{4}\right)\left(1 - \sqrt{\frac{2}{3}}\right)(f_1(u)^2 + f_1(v)^2)$$

$$+ \frac{2f_1(u)f_1(v)}{3} = -\left(\frac{\sqrt{3} - \sqrt{2}}{\sqrt{18}}\right)^2 \left(1 - \frac{\sqrt{2}}{2}\right)\frac{3 + 2\sqrt{2}}{40} - \sqrt{2}\left(1 - \sqrt{\frac{2}{3}}\right)\frac{3}{40} + \frac{2}{3}\frac{\sqrt{2}}{40}$$

$$\approx -0.0002395 - 0.0194635 + 0.0235702 \approx 0.0038672 > 0.$$

$\square$

In Table 6 we check Eldan's criterion computationally for all examples; we also check whether both our proxy estimates truthfully indicate the sign of the real spectral gap difference. Eldan's criterion $g(u, v, \cdot)$ is calculated from the sparser graph's spectral properties, as well as $\Delta$PROXYADD —estimating the spectral gap's difference when that edge is added. Meanwhile, $\Delta$PROXYDELETE is calculated from the denser graph and tries to estimate the spectral gap of the pruned one.

When $g(u, v, \cdot) > 0$, it theoretically guarantees that $\Delta\lambda_1 < 0$, i.e., that the addition of said edge is NOT desired. This holds in our table for the first and third rows, where the addition of each edge lowers the spectral gap. Our proxy values reflect it in both directions: $\Delta$PROXYADD is negative because the edge should not be added, and $\Delta$PROXYDELETE is positive because the edge should be pruned.

Note that, because of the aforementioned multiplicity of the ring's eigenvectors, if we choose another $f_1$ for the first row, Eldan's criterion might not be satisfied. For example, using the eigenvectors given by the library function $np.linalg.eigh$, the criterion yields a value of $\approx -0.005904$.

The second row shows an example of an edge that is desirable to be added. In this case, it is guaranteed that Eldan's criterion is negative. Our proxy values are again accurately descriptive of reality: $\Delta$PROXYADD is positive and $\Delta$PROXYDELETE is negative.

Table 6: Computationally calculated criteria for the toy graph examples.

| Sparser graph | Denser graph | $\{u, v\}$ | Eldan's $g(u, v, \cdot)$ | $\Delta$PROXYDELETE | $\Delta$PROXYADD | $\Delta\lambda_1$ |
|---|---|---|---|---|---|---|
| $\mathcal{G}^-$ | $\mathcal{G}$ | $\{0, 3\}$ | 0.003867 | 0.027992 | -0.017678 | -0.01002 |
| $\mathcal{G}$ | $\mathcal{G}^+$ | $\{0, 5\}$ | -0.146246 | -0.064550 | 0.415994 | 0.071632 |
| $\mathcal{G}$ | $\widetilde{\mathcal{G}^+}$ | $\{4, 7\}$ | 0.004952 | 0.032403 | -0.024739 | -0.011584 |

### A.2 Proof of Propositions 3.3 and 3.4

We choose one-dimensional features to follow normal distributions dependent on their class: $X_i \sim \mathcal{N}(1, 1)$ for class $(+)$, and $X_i \sim \mathcal{N}(-1, 1)$ for class $(-)$. After one round of mean aggregation, class $(+)$ nodes with two intra-class neighbors will still have an expected mean value of 1, because they will aggregate three features that follow the same distribution: from themselves and the two neighbors. However, nodes like $X_2$, which have one neighbor of each class, will have a lower expected value: $\frac{2-1}{3} = \frac{1}{3}$. In general, if a (class $(+)$) node has $p$ same-class neighbors and $q$ different-class neighbors, their representation after an aggregation round will follow a normal distribution $\mathcal{N}(\frac{1+p-q}{1+p+q}, \frac{1}{1+p+q})$.

The smaller its expected mean is, the more it deviates from the original mean, and the less informative it gets. In Table 7 we show the expected values of each configuration dependent on the neighbors' classes as they appear on our four considered graphs. The class $(-)$ configurations are omitted because they are the same as the ones shown but with the opposite sign.

Table 7: Expected mean values of each neighboring configuration after one round of mean aggregation.

| Name | Neighboring configuration | Expected Mean |
|------|---------------------------|---------------|
| A | | $\frac{1+1+1}{3} = 1$ |
| B | | $\frac{1+1-1}{3} = \frac{1}{3}$ |
| C | | $\frac{1+1+1-1}{4} = \frac{1}{2}$ |
| D | | $\frac{1+1-1-1}{4} = 0$ |
| E | | $\frac{1+1+1-1-1}{5} = \frac{1}{5}$ |

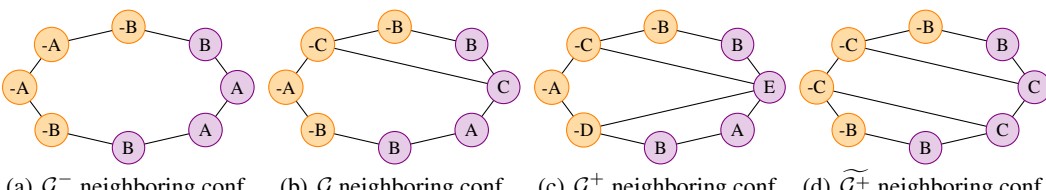

(a) $\mathcal{G}^-$ neighboring conf.   (b) $\mathcal{G}$ neighboring conf.   (c) $\mathcal{G}^+$ neighboring conf.   (d) $\widetilde{\mathcal{G}^+}$ neighboring conf.

Figure 4: Neighboring configurations on each of the four graphs from Figure 1.

Now we consider again the four graphs from Figure 1. In Figure 4 we specify which nodes have which configuration from the set {A, B, C, D, E} as named in Table 7. We arbitrarily choose orange nodes to be the negative class. After one round of mean aggregation on each of them, we can estimate the amount of class information remaining on the two classes by averaging the corresponding node representations of each node per class —that is, we average the four expected means for purple nodes and the four expected means for orange nodes. We calculate these values in Table 8. As we intended to prove, they tend towards non-informative zero for both classes as the number of edges increases, and they follow the same order as the smoothing rate curves plotted in Figure 2(a). Proposition 3.3 is proved because values tend to zero —so both classes' averages get closer together— from $\mathcal{G}^-$ to $\mathcal{G}$, and from $\mathcal{G}$ to both $\mathcal{G}^+$ and $\widetilde{\mathcal{G}^+}$. Proposition 3.4 is proved because the values from $\mathcal{G}^+$ are more informative/further apart than the values from $\widetilde{\mathcal{G}^+}$. $\qquad\square$

## B   How other label configurations affect the rings' smoothing rates

In Figure 5 we show how different configuration of labels for our example graphs affect their smoothing rate tests. In particular, we will analyze the result when added edges are intra-class instead of inter-class, as well as when the label distribution actively goes against the graph structure.

As a first modification (Figure 5(c)), we rotate the labels so that edge $\{0, 3\}$ is now intra-class; this makes edge $\{4, 7\}$ from $\widetilde{\mathcal{G}^+}$ intra-class, too. It is reflected in its smoothing rate plot in two main ways. First, the distance between graph $\mathcal{G}^-$ and $\mathcal{G}$ is not as wide, because the extra intra-edge in $\mathcal{G}$ does not cause as much smoothing as the inter-class edge from the original configuration does. Second, graph $\widetilde{\mathcal{G}^+}$ is now the least smoothed. This might be because the two edges aid in isolating the flow of information between the two, very distinct classes; note that this graph also has the smallest spectral gap, so the configuration of labels and the graph structure work towards the same goal.

Table 8: Neighboring configurations for each graph, and their average value after a round of mean aggregation.

| Node | $G^-$ | $G$ | $G^+$ | $\widetilde{G^+}$ |
|---|---|---|---|---|
| $X_6$ | B: $\frac{1}{3}$ | B: $\frac{1}{3}$ | B: $\frac{1}{3}$ | B: $\frac{1}{3}$ |
| $X_7$ | A: 1 | A: 1 | A: 1 | C: $\frac{1}{2}$ |
| $X_0$ | A: 1 | C: $\frac{1}{2}$ | E: $\frac{1}{5}$ | C: $\frac{1}{2}$ |
| $X_1$ | B: $\frac{1}{3}$ | B: $\frac{1}{3}$ | B: $\frac{1}{3}$ | B: $\frac{1}{3}$ |
| Average: | $\frac{2+\frac{2}{3}}{4} \approx 0.667$ | $\frac{1+\frac{2}{3}+\frac{1}{2}}{4} =\approx 0.542$ | $\frac{1+\frac{2}{3}+\frac{1}{5}}{4} =\approx 0.467$ | $\frac{\frac{2}{3}+\frac{2}{2}}{4} =\approx 0.417$ |
| $X_2$ | -B: $-\frac{1}{3}$ | -B: $-\frac{1}{3}$ | -B: $-\frac{1}{3}$ | -B: $-\frac{1}{3}$ |
| $X_3$ | -A: $-1$ | -C: $-\frac{1}{2}$ | -C: $-\frac{1}{2}$ | -C: $-\frac{1}{2}$ |
| $X_4$ | -A: $-1$ | -A: $-1$ | -A: $-1$ | -C: $-\frac{1}{2}$ |
| $X_5$ | -B: $-\frac{1}{3}$ | -B: $-\frac{1}{3}$ | -D: 0 | -B: $-\frac{1}{3}$ |
| Average: | $-\frac{2+\frac{2}{3}}{4} \approx -0.667$ | $-\frac{1+\frac{2}{3}+\frac{1}{2}}{4} \approx -0.542$ | $-\frac{1+\frac{1}{3}+\frac{1}{2}}{4} \approx -0.458$ | $-\frac{\frac{2}{3}+\frac{2}{2}}{4} \approx -0.417$ |

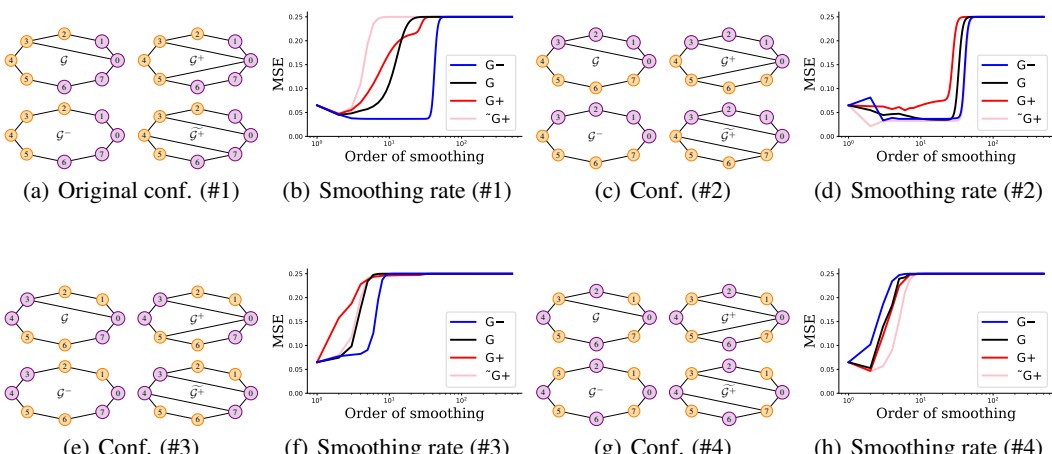

(a) Original conf. (#1)  (b) Smoothing rate (#1)  (c) Conf. (#2)  (d) Smoothing rate (#2)

(e) Conf. (#3)  (f) Smoothing rate (#3)  (g) Conf. (#4)  (h) Smoothing rate (#4)

Figure 5: Different configurations of labels/features for the example graphs of Figure 1, as well as their respective smoothing rate tests akin to Figure 2(a). Figure 5(a) is the original configuration, for direct comparison. Figure 5(c) rotates the labels and achieves more intra-class edges. Figure 5(e) achieves the same amount of intra-class edges but separates nodes with the same labels. Figure 5(g) alternates between classes and is a worse configuration to learn.

As a second modification (Figure 5(e)), we alternate classes two by two nodes at a time. This makes $\{0, 3\}$ and $\{4, 7\}$ intra-class again, so it is directly comparable to the previous disposition. However, the edges in $\widetilde{G^+}$ are not dividing the two classes so distinctively. This makes its smoothing occur more quickly than before, now on par with the base graph. We consider this phenomenon to be directly related to its lower spectral gap. Another relevant aspect of this graph is that, still, the pruned graph $G^-$ smooths less than $G$, even when the pruned edge is intra-class, and even if the spectral gap has increased; it is another instance of both mitigating over-smoothing and over-squashing.

Lastly (Figure 5(g)), we propose a configuration that is actively counterproductive to the structure of the ring, by alternating nodes of different classes one by one. As much as the spectral gap increases with the deletion of edge $\{0, 3\}$, the ring $G^-$ is a worse structure for the right kind of information to flow, and worse to avoid getting dissipated in this particular case. This unveils the ultimate limitation of not taking into account the task in a rewiring method, which is a trade-off to assume.

## C    Algorithms

Here we include the corresponding algorithms: PROXYDELETE (1), PROXYADD (2), ELDANADD (3), and ELDANDELETE (4).

---

**Algorithm 1** Proxy Spectral Gap based Greedy Graph Sparsification (PROXYDELETE)

---

**Require:** Graph $\mathcal{G} = (\mathcal{V}, \mathcal{E})$, num. edges to prune $N$, spectral gap $\lambda_1(\mathcal{L}_\mathcal{G})$, second eigenvector $f$.

  **repeat**
    **for** $(u, v) \in \mathcal{E}$ **do**
      Consider $\hat{\mathcal{G}} = \mathcal{G} \setminus (u, v)$.
      Calculate proxy value for the spectral gap of $\hat{\mathcal{G}}$ based on Eq. (1):
      $\lambda_1(\mathcal{L}_{\hat{\mathcal{G}}}) \approx \lambda_1(\mathcal{L}_\mathcal{G}) - ((f_u - f_v)^2 - \lambda_1(\mathcal{L}_\mathcal{G}) \cdot (f_u^2 + f_v^2))$
    **end for**
    Find the edge that maximizes the proxy: $(u^-, v^-) = \underset{(u,v) \in \mathcal{E}}{\arg\max}\, \lambda_1(\mathcal{L}_{\hat{\mathcal{G}}})$.
    Update graph edges: $\mathcal{E} = \mathcal{E} \setminus (u^-, v^-)$.
    Update degrees: $d_{u^-} = d_{u^-} - 1, d_{v^-} = d_{v^-} - 1$
    Update eigenvectors and eigenvalues of $\mathcal{G}$ accordingly.
  **until** $N$ edges deleted.
  **Output :** Sparse graph $\hat{\mathcal{G}} = (\mathcal{V}, \hat{\mathcal{E}})$.

---

**Algorithm 2** Proxy Spectral Gap based Greedy Graph Addition (PROXYADD)

---

**Require:** Graph $\mathcal{G} = (\mathcal{V}, \mathcal{E})$, num. edges to add $N$, spectral gap $\lambda_1(\mathcal{L}_\mathcal{G})$, second eigenvector $f$ of $\mathcal{G}$.

  **repeat**
    **for** $(u, v) \in \bar{\mathcal{E}}$ **do**
      Consider $\hat{\mathcal{G}} = \mathcal{G} \cup (u, v)$.
      Calculate proxy value for the spectral gap of $\hat{\mathcal{G}}$ based on Eq. (1):
      $\lambda_1(\mathcal{L}_{\hat{\mathcal{G}}}) \approx \lambda_1(\mathcal{L}_\mathcal{G}) + ((f_u - f_v)^2 - \lambda_1(\mathcal{L}_\mathcal{G}) \cdot (f_u^2 + f_v^2))$
    **end for**
    Find the edge that maximizes the proxy: $(u^+, v^+) = \underset{(u,v) \in \bar{\mathcal{E}}}{\arg\max}\, \lambda_1(\mathcal{L}_{\hat{\mathcal{G}}})$.
    Update graph edges: $\mathcal{E} = \mathcal{E} \cup (u^+, v^+)$.
    Update degrees: $d_{u^+} = d_{u^+} + 1, d_{v^+} = d_{v^+} + 1$
    Update eigenvectors and eigenvalues of $\mathcal{G}$ accordingly.
  **until** $N$ edges added.
  **Output :** Denser graph $\hat{\mathcal{G}} = (\mathcal{V}, \hat{\mathcal{E}})$.

---

**Algorithm 3** Eldan based Greedy Graph Addition (ELDANADD)

---

**Require:** Graph $\mathcal{G} = (\mathcal{V}, \mathcal{E})$, num. edges to add $N$, spectral gap $\lambda_1(\mathcal{L}_\mathcal{G})$, top eigenvector $f$ of $\mathcal{G}$.

  **repeat**
    **for** $edges(u, v) \in \bar{\mathcal{E}}$ **do**
      Consider $\hat{\mathcal{G}} = \mathcal{G} \cup (u, v)$.
      Compute projection $\mathcal{P}_f^2 = \langle f, \hat{f}_0 \rangle$.
      Compute Eldan's criterion $g(u, v, \mathcal{L}_\mathcal{G})$.
    **end for**
    Find the edge that minimizes the criterion: $(u^+, v^+) = \underset{(u,v) \in \bar{\mathcal{E}}}{\arg\max}\, {-}g(u, v, \mathcal{L}_\mathcal{G})$.
    $\mathcal{E} = \mathcal{E} \cup (u^+, v^+)$.
    Update degrees $d_{u^+} = d_{u^+} + 1, d_{v^+} = d_{v^+} + 1$
    Update eigenvectors and eigenvalues of $\mathcal{G}$ accordingly.
  **until** $N$ edges added.
  **Output :** Denser graph $\hat{\mathcal{G}} = (\mathcal{V}, \hat{\mathcal{E}})$.

---

**Algorithm 4** Eldan based Greedy Graph Sparsification (ELDANDELETE)

---

**Require:** Graph $\mathcal{G} = (\mathcal{V}, \mathcal{E})$, num. edges to prune $N$, spectral gap $\lambda_1(\mathcal{L}_\mathcal{G})$, top eigenvector $f$ of $\mathcal{G}$.
  **repeat**
    **for** $edges(u, v) \in \mathcal{E}$ **do**
      Consider $\hat{\mathcal{G}} = \mathcal{G} \setminus (u, v)$.
      {Note that the denser graph is the original $\mathcal{G}$, so we require approximations of $\hat{f}$ and $\lambda_1(\mathcal{L}_{\hat{\mathcal{G}}})$
      from the sparser $\hat{\mathcal{G}}$.}
      Estimate eigenvector $\hat{f}$ from $f$ based on the power iteration method.
      Estimate corresponding eigenvalue $\lambda_1(\mathcal{L}_{\hat{\mathcal{G}}})$ based on Eq. (1).
      Compute projection $\mathcal{P}_f^2 = \langle \hat{f}, f_0 \rangle$.
      Compute Eldan's criterion $g(u, v, \mathcal{L}_{\hat{\mathcal{G}}})$.
    **end for**
    Find the edge that maximizes the criterion: $(u^-, v^-) = \underset{(u,v) \in \mathcal{E}}{\operatorname{argmax}} \, g(u, v, \mathcal{L}_{\hat{\mathcal{G}}})$
    $\hat{\mathcal{E}} = \hat{\mathcal{E}} \setminus (u^-, v^-)$.
    Update degrees $d_{u^-} = d_{u^-} - 1, d_{v^-} = d_{v^-} - 1$
    Update eigenvectors and eigenvalues of $\mathcal{G}$ accordingly.
  **until** $N$ edges deleted.
  **Output :** Sparse graph $\hat{\mathcal{G}} = (\mathcal{V}, \hat{\mathcal{E}})$.

---

## D   Spectral pruning can slow down the rate of smoothing

In section §3 we have demonstrated the possibility of addressing both over-squashing and over-smoothing via spectral gap based pruning in a simple toy graph setting. Below we present the results on real-world graphs, where spectral pruning can help slow down the rate of smoothing. We adopt the same Linear GNN setup (Keriven, 2022). In Figure 6, we present two homophilic datasets (Cora and Citeseer) and two heterophilic graphs (Texas and Chameleon). For each of these experiments we add edges using FoSR (Karhadkar et al., 2023) and PROXYADD and delete edges using our proposed PROXYDELETE method. FoSR, which optimizes the spectral gap by adding edges, aids in mitigating over-squashing but inevitably leads to accelerating the smoothing rate. Conversely, if we delete edges using our PROXYDELETE method, the rate of smoothing is slowed down. It is also evident that our pruning method is more effective in heterophilous graph settings. This is likely due to the deletion of edges between nodes with different labels, thus preventing detrimental smoothing. We substantiate this by measuring the distance between nodes that have different labels, which should stay distinguishable. That is, our method deletes edges between nodes of different labels thus preventing unnecessary aggregation. We report the cosine distance for heterophilic graphs in Table 9 before training, after training on the original graph, and after training on the pruned graph. From the table it is clear that pruning edges increases the distance between nodes of different labels. Another popular metric in the literature to measure over-smoothing is Dirichlet energy, which can only measure the degree of smoothing, but not whether it is helpful for a learning task. To keep up with the trend, we plot the Dirichlet energy vs. Layers (Roth & Liebig, 2023) in Figure 7 on Cora and Texas. It is clear from the figure that our method slows down the decay of Dirichlet energy. Note that, since our method works purely on the graph topology, it cannot improve the Dirichlet energy like specialised methods (Zhou et al., 2021; Roth & Liebig, 2023; Rusch et al., 2023b).

In a recent work by Azabou et al. (2023), the authors also show similar experiments by introducing additional nodes to slow down the rate of message passing and thus slowing down the rate of smoothing. We achieve a similar effect just by pruning edges instead of introducing additional nodes.

## E   Additional results

### E.1   Node classification.

We perform semi-supervised node classification on the following datasets: Cora (McCallum et al., 2000), Citeseer (Sen et al., 2008) and Pubmed (Namata et al., 2012). We report results on Chameleon,

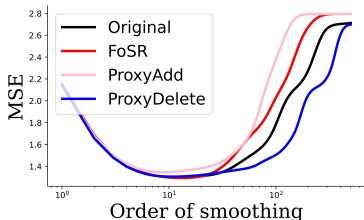

(a) Cora dataset with 200 edges added (FoSR, PROXYADD) and 20 deleted (PROXYDELETE).

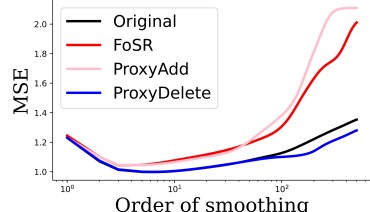

(b) Citeseer dataset with 200 edges added (FoSR, PROXYADD) and 100 deleted (PROXYDELETE).

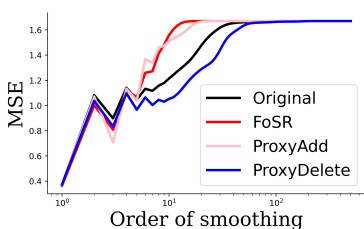

(c) Texas dataset with 200 edges added (FoSR, PROXYADD) and 5 deleted (PROXYDELETE).

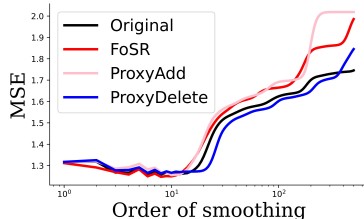

(d) Chameleon dataset with 200 edges added (FoSR, PROXYADD) and 250 deleted (PROXYDELETE).

Figure 6: We show on real-world graphs that spectral pruning can not only mitigate over-squashing by improving the spectral gap but also slows down the rate of smoothing, thus effectively preventing over-smoothing as well.

Table 9: Cosine distance between nodes of different labels before and after deleting edges using PROXYDELETE.

| Dataset | Before Training | After Training (OriginalGraph) | After Training (PrunedGraph) |
|---|---|---|---|
| Cornell | 0.72 | 0.87 | 0.83 |
| Wisconsin | 0.72 | 0.77 | 0.86 |
| Texas | 0.68 | 0.62 | 0.80 |
| Chameleon | 0.99 | 0.91 | 0.96 |
| Squirrel | 0.98 | 0.82 | 0.89 |
| Actor | 0.83 | 0.95 | 0.99 |

Squirrel, Actor and the WebKB datasets consisting of Cornell, Wisconsin and Texas. Our baselines include GCN (Kipf & Welling, 2017) without any modifications to the original graph, DIGL by Gasteiger et al. (2019), SDRF by Topping et al. (2022), and FoSR by Karhadkar et al. (2023). We adopt the public implementations available and tune the hyperparameters to improve the performance if possible. Our results are presented in Table 10. We compare GCN with no edge modifications, GCN+DIGL, GCN+SDRF, GCN+FoSR, GCN+RandomDelete, GCN+ELDANDELETE where we delete the edges, GCN+ELDANADD where we add the edges according to the criterion from Lemma 3.1 and PROXYADD and PROXYDELETE which use Equation (1) to optimize the spectral gap directly. The results for GCN+BORF (Nguyen et al., 2023) are taken from the paper directly, hence NA for some datasets. The top performance is highlighted in **bold**. GCN+FoSR outperforms all methods on Cora, Citeseer and Pubmed, which are homophilic. Yet, GCN+PROXYADD is more effective in increasing the spectral gap (see Table 16). On the remaining six datasets, our proposed methods both with edge deletions and additions outperform FoSR and SDRF, while outperforming all other baselines on all datasets. For training details and hyperparameters, please refer to the Appendix 19.

### E.2 Graph classification with GCN and R-GCN

We conduct experiments on graph classification with a GCN (Kipf & Welling, 2017) and R-GCN (Battaglia et al., 2018) backbone to demonstrate the effectiveness of our proposed rewiring algorithms.

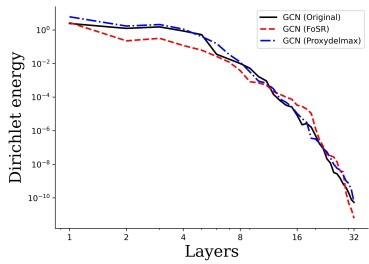
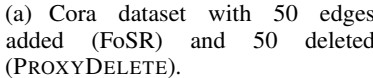

(a) Cora dataset with 50 edges added (FoSR) and 50 deleted (PROXYDELETE).

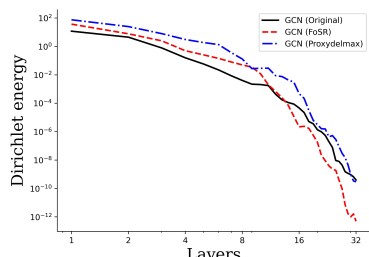

(b) Texas dataset with 50 edges added (FoSR) and 100 deleted (PROXYDELETE) .

Figure 7: We measure the Dirichlet energy and plot it for increasing depth on a homophilic dataset, Cora and a heterophilic dataset, Texas. For increasing depth, we can see PROXYDELMAX slows the decay of Dirichlet energy.

Table 10: We compare the performance of GCN augmented with different graph rewiring methods on node classification.

| Method | Cora $\mathcal{H} = 0.8041$ | Citeseer $\mathcal{H} = 0.7347$ | Pubmed $\mathcal{H} = 0.8023$ | Cornell $\mathcal{H} = 0.1227$ | Wisconsin $\mathcal{H} = 0.1777$ | Texas $\mathcal{H} = 0.060$ | Actor $\mathcal{H} = 0.2167$ | Chameleon $\mathcal{H} = 0.2474$ | Squirrel $\mathcal{H} = 0.2174$ |
|---|---|---|---|---|---|---|---|---|---|
| GCN | 87.22±0.40 | 77.35±0.70 | 86.96±0.17 | 50.74±1.63 | 53.52±1.50 | 50.40±1.47 | 29.12±0.24 | 31.15±0.84 | 26.00±0.69 |
| GCN+DIGL | 83.21±0.79 | 73.29±0.17 | 78.84±0.008 | 42.04±4.43 | 44.22±5.02 | 57.35±6.46 | 26.33±1.22 | 38.95±0.99 | 32.45±0.88 |
| GCN+SDRF | 87.84±0.68 | 78.43±0.62 | 87.36±0.14 | 53.54±2.65 | 58.78±3.22 | 60.25±4.97 | 31.67±0.36 | 41.30±1.36 | 38.98±0.46 |
| GCN+FoSR | **91.44±0.39** | **82.13±0.31** | **91.49±0.10** | 53.91±1.47 | 58.63±1.46 | 63.50±1.75 | 38.01±0.21 | 46.64±0.63 | 50.73±0.37 |
| GCN+ELDANDELETE | 87.60±0.18 | 78.68±0.54 | 87.33±0.07 | 65.13±1.50 | **67.84±1.45** | 70.53±1.23 | **43.65±0.21** | 52.51±0.55 | 48.89±0.40 |
| GCN+ELDANADD | 88.38±0.12 | 79.45 ±0.37 | 87.17±0.14 | **69.05±1.50** | 64.08±1.63 | 67.10±1.13 | 43.64±0.25 | 48.09±0.59 | **51.66±0.45** |
| GCN+PROXYADD | 89.10±0.70 | 78.94±0.54 | 87.54±0.24 | 66.54±1.41 | 67.75±1.64 | **74.21±1.25** | 43.45±0.20 | 54.30±0.59 | 48.85±0.39 |
| GCN+PROXYDELETE | 87.51±0.81 | 78.68 ±0.55 | 87.39±0.11 | 66.60 ± 1.67 | 66.36±1.33 | 72.36±1.35 | 43.52±0.22 | 55.88±0.70 | 48.90±0.39 |
| GCN+RANDOMDELETE | 87.30±0.31 | 78.34±0.38 | 87.15±0.16 | 63.97±2.50 | 61.71±2.73 | 63.97±5.41 | 29.57±0.44 | 44.07±1.04 | 40.63±0.41 |
| GCN+BORF | 87.50±0.20 | 73.80±0.20 | NA | 50.80±1.11 | 50.30±0.90 | 49.40±1.20 | NA | **61.50±0.40** | NA |

Our experimental setting is the same as that of FoSR by Karhadkar et al. (2023), with the difference being we tune our hyperparameters on 10 random splits instead of 100. The final test accuracy is averaged over 5 random splits of data. We compare our results with FoSR by Karhadkar et al. (2023). For the IMDB-BINARY, REDDIT-BINARY and COLLAB datasets there are no node features available and have to be created. For fair comparison we run FoSR on these datasets. For ENZYMES and MUTAG the results are taken from the values reported in the paper. The results are reported in Table 11 and 12. From the tables it is clear that our proposed algorithms are effective in increasing the generalization performance even for graph classification tasks.

Table 11: Graph classification with GCN comparing FoSR, ELDANADD and PROXYADD.

| Method | ENZYMES | MUTAG | IMDB-BINARY | REDDIT-BINARY | COLLAB | PROTEINS |
|---|---|---|---|---|---|---|
| GCN+FoSR | 25.06±0.50 | 80.00±0.80 | 68.80±4.04 | 80.01±0.02 | 80.30±0.00 | 73.42 ± 0.41 |
| GCN+ELDANADD | 26.36±0.01 | 82.16±0.03 | **75.84±0.01** | **81.03±0.02** | **81.82±0.97** | 70.53±0.86 |
| GCN+PROXYADD | **27.39±0.01** | **85.00±0.00** | 75.00±0.02 | 78.20±0.01 | 79.52±0.01 | **76.53±0.02** |

Table 12: Graph classification with R-GCN comparing FoSR, ELDANADD and PROXYADD.

| Method | ENZYMES | MUTAG | IMDB-BINARY | REDDIT-BINARY | COLLAB | PROTEINS |
|---|---|---|---|---|---|---|
| R-GCN+FoSR | **35.63±0.58** | 84.45±0.77 | 70.16±3.67 | 80.01±0.02 | 78.04±0.84 | **73.79±0.35** |
| R-GCN+ELDANADD | 30.55±0.16 | **85.80±0.20** | **76.32±0.07** | 79.76±0.17 | 80.69±0.01 | 72.01±0.04 |
| R-GCN+PROXYADD | 33.12±2.74 | 78.0±5.51 | 73.96±2.25 | **87.93±0.61** | **80.22±1.13** | 73.32±2.78 |

## E.3 Node classification using Relational-GCN

In Table 13 we compare FoSR (Karhadkar et al., 2023) and our proposed methods that use Eldan's criterion for adding edges and the PROXYADD method with a Relational-GCN backbone on 9 datasets.

We adopt the experimental setup and code base of (Karhadkar et al., 2023), with the exception of averaging over 10 random splits of data instead of 100.

Table 13: Node classification using Relational-GCNs comparing FoSR, Eldan's criterion and PROXYADD.

| Method | Cora | Citeseer | Pubmed | Cornell | Wisconsin | Texas | Actor | Chameleon | Squirrel |
|---|---|---|---|---|---|---|---|---|---|
| R-GCN+FoSR | 87.28±0.67 | 73.81±0.10 | 88.61±0.28 | 71.62±2.88 | 76.07±5.16 | 75.40±3.77 | **35.19±0.49** | 39.83±2.70 | **34.80±1.34** |
| R-GCN+ELDANADD | 87.38±1.03 | 73.72±1.15 | 88.58±0.20 | **73.78±6.30** | **77.45±3.19** | **78.37±2.75** | 34.75±0.40 | **43.20±1.24** | 33.79±0.81 |
| R-GCN+PROXYADD | **87.42±0.01** | **75.82±0.09** | **89.17 ± 0.42** | 70.00±0.20 | **77.45±0.40** | 75.67±0.40 | 35.05±0.35 | 42.58±1.20 | 33.03±1.40 |

# F  Update period, empirical runtimes and spectral gap comparisons

In §4.1, we have discussed the time complexity analysis of our proposed algorithms. Recall, that our algorithm has a hyperparameter $M$, the number of edges to delete after ranking the edges using our proxy. For edge additions, the candidate edges that can be added are large, thus we can resort to sampling a constant set of edges to speed up the process. All of our experiments in §5 were conducted with $M = 1$. However, it is possible to further reduce the overall runtimes by tuning the value of $M$, that is, how many edges we can modify before we have to recalculate the proxy to rank the edges again. This is shown in Table 15, where we compare our algorithms with $M = 1$ and $M = 10$, for 50 edge modifications. It is clear that although $M = 1$ leads to better spectral gap improvement, $M = 10$ is also a valid updating period which induces enough spectral gap change while simultaneously bringing down the runtime (also presented in Table 14) considerably, especially for large graphs. To further evaluate the trade-off between the update period and its effect on GNN test accuracy, we use PROXYADD and PROXYDELETE with different $M$ updates on Cora and Texas datasets to modify 50 and 20 edges respectively. This is shown in Figure 8. Although a more frequent update points to better test accuracy, update periods with $\{5, 10\}$ also yield competitive results. Thus reinforcing the fact that our proposed methods can be computationally efficient and can help in improving the generalization. In Table 16 we report the spectral gap changes induced by FoSR (Karhadkar et al., 2023), our proposed Eldan criterion based addition and deletions and also the Proxy versions of addition and deletions. In Table 17 we provide the runtimes for large heterophilic datasets (Platonov et al., 2023).

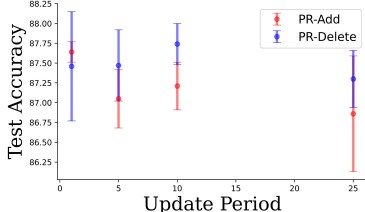

(a) Test accuracy for Cora with $M = \{1, 5, 10, 25\}$ update periods. We add/delete 50 edges.

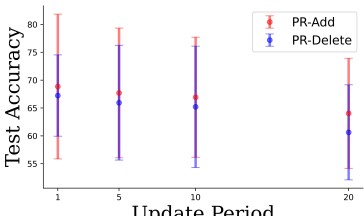

(b) Test accuracy for Texas with $M = \{1, 5, 10, 20\}$ update periods. We add/delete 20 edges.

Figure 8: We investigate the trade-off between how frequently we need to update the ranking criterion vs. the test accuracy for GCN on Cora and Texas for node classification.

Table 14: Runtimes for 50 edge modifications in seconds.

| Method | Cora | Citeseer | Chameleon | Squirrel |
|---|---|---|---|---|
| FoSR | 4.69 | 5.33 | 5.04 | 19.48 |
| SDRF | 19.63 | 173.92 | 17.93 | 155.95 |
| PROXYADD | 4.30 | 3.13 | 1.15 | 9.12 |
| PROXYDELETE | 1.18 | 0.86 | 1.46 | 7.26 |

Table 15: Empirical runtime (RT) comparisons with different update periods for the criterion for 50 edges. We also report the spectral gap before (SG.B) and after rewiring (SG.A).

| Method | Cora | | | Citeseer | | | Chameleon | | | Squirrel | | |
|---|---|---|---|---|---|---|---|---|---|---|---|---|
| Measures | SG.B | SG.A | RT | SG.B | SG.A | RT | SG.B | SG.A | RT | SG.B | SG.A | RT |
| PROXYADD (M=1) | 0.00478 | 0.0240 | 41.82 | 0.0015 | 0.012 | 27.70 | 0.0063 | 0.018 | 9.24 | 0.051 | 0.069 | 75.89 |
| PROXYDELETE (M=1) | 0.00478 | 0.0059 | 12.82 | 0.0015 | 0.0018 | 5.47 | 0.0063 | 0.0064 | 7.51 | 0.051 | 0.053 | 66.00 |
| PROXYADD (M=10) | 0.00478 | 0.018 | 4.30 | 0.0015 | 0.0067 | 3.13 | 0.0063 | 0.0160 | 1.15 | 0.051 | 0.058 | 9.12 |
| PROXYDELETE (M=10) | 0.00478 | 0.0074 | 1.04 | 0.0015 | 0.0021 | 0.86 | 0.0063 | 0.0065 | 1.46 | 0.051 | 0.0527 | 7.26 |

Table 16: We compare the spectral gap improvements of different rewiring methods for 50 edge modifications. From the table it is evident that our proposed PROXYADD and PROXYDELETE methods improve the spectral gap much better than FoSR.

| Method | Cora | | Citeseer | | Chameleon | | Squirrel | |
|---|---|---|---|---|---|---|---|---|
| Spectral Gap Changes | SG. Before | SG. After | SG. Before | SG. After | SG. Before | SG. After | SG. Before | SG. After |
| FoSR | 0.0047 | 0.0099 | 0.0015 | 0.0027 | 0.0063 | 0.0085 | 0.051 | 0.052 |
| PROXYADD | 0.0047 | 0.024 | 0.0015 | 0.012 | 0.0063 | 0.018 | 0.051 | 0.069 |
| PROXYDELETE | 0.0047 | 0.0059 | 0.0015 | 0.0018 | 0.0063 | 0.0064 | 0.051 | 0.053 |
| ELDANADD | 0.0047 | 0.0047 | 0.0015 | 0.0039 | 0.0063 | 0.0085 | 0.051 | 0.052 |
| ELDANDELETE | 0.0047 | 0.0074 | 0.0015 | 0.0099 | 0.0063 | 0.0059 | 0.051 | 0.053 |

Table 17: Spectral gap changes and empirical runtimes for Large Heterophilic Datasets.

| Dataset | SG. Before | #EdgesAdded | SG. After | AddingTime | #EdgesDeleted | SG. After | PruningTime |
|---|---|---|---|---|---|---|---|
| Roman-Empire | 3.6842e-07 | 5 | 7.5931e-07 | 124.34 | 20 | 3.6875e-07 | 9.23 |
| Amazon-Ratings | 0.000104825 | 10 | 0.000230704 | 380.62 | 50 | 0.000104840 | 62.69 |
| Minesweeper | 0.000376141 | 20 | 0.000375844 | 164.44 | 20 | 0.000376141 | 15.5 |

# G   Pruning at initialization for graph lottery tickets

In Table 18, we present the results for Pruning at Initialization for finding graph lottery tickets. We first prune the input graph to the required sparsity level and then the weights are iteratively pruned by magnitude similar to the approach proposed by (Chen et al., 2021). From the table it is clear that, at least for moderate graph sparsity (GS) levels for Cora dataset, that is around GS = $18.75\%$, our proposed ELDANDELETE-UGS and PROXYDELETE attain comparable performance to UGS. On Pubmed for different graph sparsity levels we outperform UGS. Meanwhile, our method fails to identify winning tickets for Citeseer. We use the public implementation by the authors (Chen et al., 2021) for all our lottery ticket experiments. For all experiments we report the test accuracy on node classification averaged over 3 runs. Except for Pubmed which could only be averaged over 2 runs.

Table 18: We perform pruning at initialization to find graph lottery tickets. We compare UGS with our proposed methods for varying graph sparsity (GS) and weight sparsity (WS) levels.

| Cora - GS(18.75%); WS(89.88%) | | | Citeseer - GS(19.46%);WS(89.80%) | | | Pubmed - GS(19.01%);WS(89.33%) | |
|---|---|---|---|---|---|---|---|
| Method | Acccuracy | | Method | Accuracy | | Method | Accuracy |
| UGS | 79.54±1.20 | | UGS | 72.20±0.60 | | UGS | 77.75±1.04 |
| Eldan-UGS | 79.10±0.07 | | Eldan-UGS | 68.15±0.65 | | Eldan-UGS | 79.80±0.00 |
| ProxyDelete-UGS | 78.66±0.73 | | PROXYDELETE-UGS | 69.76±0.65 | | ProxyDelete-UGS | 78.20±0.20 |

| Cora - GS(57.59%) WS(98.31%) | | | Citeseer- GS(59.12%);WS(98.12%) | | | Pubmed - GS(56.47%);WS(98.21%) | |
|---|---|---|---|---|---|---|---|
| UGS | 72.65±0.55 | | UGS | 68.70±0.20 | | UGS | 76.80±0.00 |
| Eldan-UGS | 72.40±0.40 | | Eldan-UGS | 66.55±0.15 | | Eldan-UGS | 77.70±0.00 |
| ProxyDelete-UGS | 70.49±0.27 | | PROXYDELETE-UGS | 67.96±1.72 | | ProxyDelete-UGS | 77.80±0.00 |

| Cora - GS(78.81%) WS(98.23%) | | | Citeseer- GS(82.63%);WS(98.59%) | | | Pubmed - GS(81.01%);WS(97.19%) | |
|---|---|---|---|---|---|---|---|
| UGS | 68.65±0.95 | | UGS | 66.05±0.45 | | UGS | 76.25±0.45 |
| Eldan-UGS | 67.20±0.10 | | Eldan-UGS | 62.60±0.60 | | Eldan-UGS | 72.80±0.00 |
| ProxyDelete-UGS | 64.46±0.47 | | PROXYDELETE-UGS | 61.19±0.29 | | ProxyDelete-UGS | 74.70±0.00 |

# H   Training details and hyperparameters

We instantiate a 2-layered GCN (Kipf & Welling, 2017) for semi-supervised node classification, the Planetoid datasets (Cora, Citeseer and Pubmed) are available as pytorch geometric datasets. For the WebKB datasets we use the updated ones given by Platonov et al. (2023). We use a 60/20/20 split for training/testing/validation respectively for all datasets. We perform extensive hyperparameter tuning on the validation set and finally report test accuracy averaged over 10 splits of the data (Chen et al., 2018). We use the largest connected component wherever available. The same experimental settings hold for other baselines DIGL, SDRF and FoSR. For node classification using R-GCNs, we also use a 3 layered GCN, this is highlighted in Table 20 with other hyperparameters. For graph classification, we use the same experimental setup as (Karhadkar et al., 2023), we use a 4-layered GCN and R-GCN versions. For the larger heterophilic datasets, we use the experimental setup given by the authors (Platonov et al., 2023). We set the learning rate to $\{3e-3, 3e-4\}$, dropout to $0.32$, and the hidden dimension size to $512$. For GATs, the attention heads are set to $8$. The datasets are split into 50%/25%/25% for train, test and validation respectively. We tune our edge modification algorithms on the validation set. The final test accuracy is reported as averaged over 10 random splits run for 1000 steps. Skip connections and normalization (Ba et al., 2016) is used to facilitate training deeper models. We use PyTorch Geometric and DGL library for our experiments. All experiments were done on 2 V100 GPUs. Our code `https://github.com/RelationalML/SpectralPruningBraess` is available.

Table 19: Hyperparameters for GCN+our proposed rewiring algorithms.

| Dataset | LR | HiddenDimension | Dropout | ELDANADD | ELDANDELETE | PROXYADD | PROXYDELETE |
|---------|-----|-----------------|-----------|----------|-------------|----------|-------------|
| Cora | 0.01 | 32 | 0.3130296 | 50 | 20 | 100 | 100 |
| Citeseer | 0.01 | 32 | 0.4130296 | 50 | 20 | 50 | 50 |
| Pubmed | 0.01 | 128 | 0.3130296 | 50 | 100 | 20 | 50 |
| Cornell | 0.001 | 128 | 0.4130296 | 100 | 5 | 50 | 20 |
| Wisconsin | 0.001 | 128 | 0.5130296 | 100 | 5 | 50 | 10 |
| Texas | 0.001 | 128 | 0.4130296 | 100 | 5 | 50 | 76 |
| Actor | 0.001 | 128 | 0.2130296 | 100 | 10 | 25 | 500 |
| Chameleon | 0.001 | 128 | 0.2130296 | 100 | 50 | 50 | 200 |
| Squirrel | 0.001 | 128 | 0.5130296 | 50 | 100 | 10 | 1000 |

Table 20: Hyperparameters for R-GCN+PROXYADD on node classification

| Dataset | LR | Layers | HiddenDimension | Dropout | PROXYADD |
|---------|-----|--------|-----------------|-----------|----------|
| Cora | 0.01 | 2 | 32 | 0.3130296 | 50 |
| Citeseer | 0.01 | 2 | 64 | 0.3130296 | 250 |
| Pubmed | 0.01 | 2 | 32 | 0.4130296 | 100 |
| Cornell | 0.001 | 3 | 128 | 0.3130296 | 05 |
| Wisconsin | 0.001 | 3 | 128 | 0.3130296 | 25 |
| Texas | 0.01 | 3 | 128 | 0.3130296 | 20 |
| Actor | 0.001 | 3 | 128 | 0.5130296 | 25 |
| Chameleon | 0.001 | 3 | 128 | 0.4130296 | 100 |
| Squirrel | 0.001 | 3 | 128 | 0.3130296 | 5 |

Table 21: Hyperparameters for graph classification with GCN+EldanAdd

| Dataset | LR | Dropout | Hidden Dimension | EldanAdd |
|---------|-----|---------|------------------|----------|
| ENZYMES | 0.001 | 0.2130296 | 32 | 20 |
| MUTAG | 0.001 | 0.3130296 | 32 | 20 |
| IMDB-BINARY | 0.001 | 0.3130296 | 32 | 10 |
| REDDIT-BINARY | 0.001 | 0.2130296 | 32 | 10 |
| COLLAB | 0.001 | 0.21 | 32 | 10 |
| PROTEINS | 0.001 | 0.3130296 | 32 | 10 |

Table 22: Hyperparameters for graph classification with GCN + PROXYADD

| Dataset | LR | Dropout | Hidden Dimension | ProxyAdd |
|---------|-----|---------|------------------|----------|
| ENZYMES | 0.001 | 0.2130296 | 32 | 20 |
| MUTAG | 0.001 | 0.3130296 | 32 | 20 |
| IMDB-BINARY | 0.001 | 0.3130296 | 32 | 10 |
| REDDIT-BINARY | 0.001 | 0.2130296 | 32 | 10 |
| COLLAB | 0.001 | 0.21 | 32 | 10 |
| PROTEINS | 0.001 | 0.3130296 | 32 | 10 |

Table 23: Hyperparameters for SDRF.

| Dataset | LR | Dropout | Hidden Dimension | SDRF Iterations | $\tau$ | $C^+$ |
|---------|-----|---------|------------------|-----------------|--------|-------|
| Cora | 0.01 | 0.3130296 | 32 | 100 | 163 | 0.95 |
| Citeseer | 0.01 | 0.2130296 | 32 | 84 | 180 | 0.22 |
| Pubmed | 0.01 | 0.4130296 | 128 | 166 | 115 | 1443 |
| Cornell | 0.001 | 0.2130296 | 128 | 126 | 145 | 0.88 |
| Wisconsin | 0.001 | 0.2130296 | 128 | 89 | 22 | 1.64 |
| Texas | 0.001 | 0.2130296 | 128 | 136 | 12 | 7.95 |
| Actor | 0.01 | 0.4130296 | 128 | 3249 | 106 | 7.91 |
| Chameleon | 0.01 | 0.2130296 | 128 | 2441 | 252 | 2.84 |
| Squirrel | 0.01 | 0.2130296 | 128 | 1396 | 436 | 5.88 |

Table 24: Hyperparameters for FoSR.

| Dataset | LR | Dropout | Hidden Dimension | FoSR Iterations |
|---------|-----|---------|------------------|-----------------|
| Cora | 0.01 | 0.5130296 | 128 | 50 |
| Citeseer | 0.01 | 0.3130296 | 128 | 10 |
| Pubmed | 0.01 | 0.4130296 | 128 | 50 |
| Cornell | 0.001 | 0.2130296 | 128 | 100 |
| Wisconsin | 0.001 | 0.2130296 | 128 | 100 |
| Texas | 0.001 | 0.4130296 | 128 | 100 |
| Actor | 0.01 | 0.4130296 | 128 | 100 |
| Chameleon | 0.01 | 0.4130296 | 128 | 100 |
| Squirrel | 0.01 | 0.2130296 | 128 | 100 |

Table 25: Hyperparameters for DIGL.

| Dataset | LR | Dropout | Hidden Dimension | $\alpha$ | $\kappa$ |
|---------|-----|---------|------------------|----------|----------|
| Cora | 0.01 | 0.41 | 32 | 0.0773 | 128 |
| Citeseer | 0.01 | 0.31 | 32 | 0.1076 | - |
| Pubmed | 0.01 | 0.41 | 128 | 0.1155 | 128 |
| Cornell | 0.001 | 0.41 | 128 | 0.1795 | 64 |
| Wisconsin | 0.001 | 0.31 | 128 | 0.1246 | - |
| Texas | 0.001 | 0.41 | 128 | 0.0206 | 32 |
| Actor | 0.01 | 0.21 | 128 | 0.0656 | - |
| Chameleon | 0.01 | 0.41 | 128 | 0.0244 | 64 |
| Squirrel | 0.01 | 0.41 | 128 | 0.0395 | 32 |

Table 26: Hyperparameters for graph classification with R-GCN+EldanAdd

| Dataset | LR | Dropout | Hidden Dimension | EldanAdd |
|---------|-----|---------|------------------|----------|
| ENZYMES | 0.001 | 0.2130296 | 64 | 50 |
| MUTAG | 0.001 | 0.3130296 | 64 | 40 |
| IMDB-BINARY | 0.001 | 0.3130296 | 32 | 50 |
| REDDIT-BINARY | 0.001 | 0.2130296 | 32 | 50 |
| COLLAB | 0.01 | 0.4130296 | 32 | 05 |
| PROTEINS | 0.01 | 0.4130296 | 32 | 05 |

Table 27: Hyperparameters for graph classification with R-GCN + PROXYADD

| Dataset | LR | Dropout | Hidden Dimension | ProxyAdd |
|---------|-----|---------|------------------|----------|
| ENZYMES | 0.001 | 0.2130296 | 32 | 10 |
| MUTAG | 0.001 | 0.3130296 | 32 | 10 |
| IMDB-BINARY | 0.001 | 0.3130296 | 32 | 10 |
| REDDIT-BINARY | 0.001 | 0.2130296 | 32 | 20 |
| COLLAB | 0.01 | 0.4130296 | 32 | 10 |
| PROTEINS | 0.01 | 0.4130296 | 32 | 10 |

