# OpenReview forum: "Spectral Graph Pruning Against Over-Squashing and Over-Smoothing"
_NeurIPS.cc/2024/Conference — NeurIPS 2024 poster_

### Official Review · Reviewer_MxtA · 2024-06-24

**Soundness:** 3
**Presentation:** 3
**Contribution:** 2
**Rating:** 6
**Confidence:** 3

**Summary:**

In this paper, the authors propose several variations of a graph pruning/rewiring algorithm, based either on an approximate maximization of the spectral gap, or on a more complex criterion based on Eldan's proof that deleting edges can counter-intuitively lead to an increasing spectral gap (which has been linked to the so-called Braess' paradox). They claim that, by maximizing the spectral gap, their algorithm fights over-squashing, while deleting edges naturally leads to combatting oversmoothing at the same time. They prove these properties on a toy example ring graph. They then show improvements when training GNNs on pruned graphs, mostly on heterophilic graphs, and also gives connections with the recent trend of finding "lottery tickets" in graphs.

**Strengths:**

- the paper creates connections between different literatures, all important in their own rights
- the paper is well-written and articulated
- the experiments are quite complete, with additional results in appendix

**Weaknesses:**

- some "connections" claimed by the authors appear quite tenuous or unclear, eg the graph "lottery tickets" seems a bit unnecessary and not really motivated (experiments are indeed interesting, but I would not say that playing with the spectral gap, a very natural and extensively studied idea, "creates a connection with graph lottery tickets")

- my main comment is that, although the spectral gap is indeed used as a proxy for over-squashing, this is often motivated by Cheeger's inequality (as described by the authors), as the Cheeger's constant is admitted to be the "real" measure of oversquashing. However, unlike the spectral gap, it is easy to show that Cheeger's constant is *always decreasing with edges deletion*. Hence the claim that there may be a "paradox" that leads to reducing both over-squashing and oversmoothing at the same time should be quite toned down in my opinion.

**Questions:**

- Although the algorithm seems technically novel, the idea of maximizing the spectral gap has been extensively explored in various processes on graphs, and indeed in the vast literature of graph sparsification. Is there a connection between your approach and previous approaches for sparsifying graphs, see eg "Spectral sparsification of graphs: theory and algorithms" by Batson et al.

- for graph lottery, are the "-UGS" versions of the algorithm differents ? It is never clearly explained.

**Limitations:**

See above

---

> ### Author Rebuttal · Authors · 2024-08-06
>
> We thank reviewer MxtA for the constructive feedback provided and valuable insights. We include answers for each of the points raised.
>
> * W1. We believe the idea of graph lottery tickets is an interesting application since we propose a graph sparsification method. Recent works [Pal, Hoang] suggest the importance of preserving the spectral properties (Ramanujan graph property) during finding lottery tickets and our proposed approach complements this line of work (L325 in our paper).
>
> * W2. We agree and would be happy to add a more nuanced discussion in a revision. Our position is that quantifying the phenomenon of over-squashing is an active area of research [DiGiovanni], and, while various metrics have been proposed, the spectral gap has an intuitive explanation in terms of random walks. It is also reasonable to think that any good over-squashing definition would exhibit a parallel Braess phenomenon in effect. Furthermore, our results on the Long Range Graph Benchmark [Dwivedi] (Table 1 of our paper and also Table 5 in the attachment where we have also added FoSR [Karhadkar] as a new baseline) suggest that our rewiring strategy are able to mitigate over-squashing.
>
> * Q1. We would like to thank the reviewer for bringing this interesting work to our attention. Our goal in the paper is to obtain a sparse representation of the graph by deleting edges that maximize the spectral gap in an efficient way. The methods mentioned in [Batson] point to finding sparse representations of graphs that preserve the spectral properties of the original graph, which we would be happy to mention in a revision.
>
> * Q2. For finding graph lottery tickets we use the algorithms provided by UGS [Chen] with a notable difference: we adopt a spectral gap based pruning for the input graph but use the original iterative magnitude pruning (IMP) for the network weights, while UGS uses iterative magnitude pruning for both the input graph and the network weights.
>
> References:
> * [Pal] Pal. et al. A study on the ramanujan graph property of winning lottery tickets. ICML 2022.
> * [Hoang] Hoang et al. Revisiting pruning at initialization through the lens of Ramanujan graph. ICLR 2023
> * [Batson] Batson et al. Spectral sparsification of graphs: theory and algorithms. Comms ACM. 2013.
> * [Chen] Chen et al. A Unified Lottery Ticket Hypothesis for Graph Neural Networks. ICML. 2021
> * [DiGiovanni] Di Giovanni et al. How does over-squashing affect the power of GNNs? TMLR 2024.
> * [Gutteridge] Gutteridge. et al . DRew: Dynamically rewired message passing with delay, ICML 2023.
> * [Dwivedi] Dwivedi. et al. Long range graph benchmark, NeurIPS 2022.
> * [Karhadkar] Karhadkar et al. FoSR: First-order spectral rewiring for addressing oversquashing in GNNs. ICLR 2023.

---

> > ### Comment · Reviewer_MxtA · 2024-08-12
> >
> > I thank the authors for their answers, which address most of my questions. I still find the lottery tickets to be a bit tangential and the ``paradox'' to be a bit ``on-the-nose'' for something as studied as spectral gap maximization, but admit that this is an interesting work that draws interesting connections with GNN literature. I will keep my score as is, but tentatively recommend acceptance.

---

### Official Review · Reviewer_1jBY · 2024-07-01

**Soundness:** 3
**Presentation:** 3
**Contribution:** 3
**Rating:** 6
**Confidence:** 3

**Summary:**

This paper investigates the connection between oversmoothing/oversquashing, spectral gap optimization via edge deletions and additions, and the lottery ticket hypothesis. Specifically, they propose that sparsifying a graph can indeed improve its response to oversmoothing / oversquashing. Some theoretical studies is done over rings, and empirical results are given that show promising results in practice.

**Strengths:**

I thought this paper was really interesting, and takes an ambitious step to try to tie these topics together. Given that connecting foundational graph theoretical concepts with message passing in machine learning is still tenuous, I think the discussion in this paper is interesting and novel.

**Weaknesses:**

I think depending too much on rings is a bit too idiosyncratic; there's no reason to assume that most graphs in practice are ring-like, and the circulant structure is too specific. That being said, the numerical results are done over more general classes of graphs and still seem promising, so the general intuition is probably correct.

The condition in lemma 3.1 is a bit too specific Can you give some intuition as to what we should interpret from it?

**Questions:**

Is it obvious that increasing the spectral graph is always good for message passing? First, the smallest nonzero eigenvalue only discusses the first partition, but if the task is multiclass classification, or regression, then the next smallest eigenvalues should also play a role. More generally, it seems that relying too much on Cheeger's inequality isn't precisely telling the whole picture; do you have thoughts as to whether a larger part of the spectrum might be used in this analysis?

**Limitations:**

none exist

---

> ### Author Rebuttal · Authors · 2024-08-06
>
> We thank reviewer 1jBY for the constructive feedback provided and valuable insights. We include answers for each of the points raised.
> * W1. The ring graph has the specific purpose of providing a counterexample for the hypothesis that oversmoothing and oversquashing have to be traded-off against each other, which appears in previous analyses of the over-squashing literature [Karhadkar, Giraldo, Banerjee]. It also provides an intuition to the reader as to when the Braess phenomenon can be exploited to alleviate these two problems simultaneously. Our experiments on real-world graphs indeed demonstrate that our theoretical insights can be exploited and carried over into practice.
> * W2. [Eldan] gives a concise explanation of their lemma in mathematical terms in their section 1.3: “Given a general graph G, and another graph G+ obtained from G by adding a single edge, consider the second eigenvector v2 of the normalized Laplacian LG, and let v’2 be the projection of v2 away from the top eigenvector of LG+. If v’2 has a smaller Rayleigh quotient in G+ than in G, then the spectral gap decreases. This event can be explicitly expressed using v2, λ2(LG), and the degrees of vertices in G [...].” In the context of graph rewiring, it offers us a conservative criterion that even provides a guarantee that an edge deletion will lead to a spectral gap increase. We use the stated quantity to rank which edges to delete.
> * Q1. Controlling the full spectrum during rewiring is a very interesting idea. However, when one analyzes graphs with respect to random walks, the dominant factor in their convergence is the first eigenvalue. Smaller eigenvalues might play a role if the corresponding eigen directions are pronounced in the initial conditions. We consider this somewhat unlikely during general message passing. A different kind of problem would be to study the condition number of the graph; that is, the ratio between smallest and largest eigenvalue. In terms of how hard the learning problem is, the condition number could be important in a future analysis. However, this analysis would probably address trainability more rather than oversquashing or oversmoothing.
>
> References:
> * [Karhadkar] Karhadkar et al. FoSR: First-order spectral rewiring for addressing oversquashing in GNNs. ICLR 2023.
> * [Giraldo] Giraldo et al. On the Trade-off between Over-smoothing and Over-squashing in Deep Graph Neural Networks. ACM CIKM 2023.
> * [Banerjee] Banerjee et al. Oversquashing in GNNs through the lens of information contraction and graph expansion. In 2022 58th Annual Allerton Conference on Communication, Control, and Computing (Allerton).
> * [Eldan] Eldan et al. Braess's paradox for the spectral gap in random graphs and delocalization of eigenvectors. Random Struct. Algorithms 2015.

---

> > ### Comment · Reviewer_1jBY · 2024-08-10
> >
> > Re W1, W2, ok, fair enough if the purpose is to give a small example to break a common misconception. It does limit the broader applicability a bit, and maybe some efforts should be made there, e.g. find some random graph examples where this is true numerically.
> >
> > Re Q1: I think this strikes the heart of the difference between optimizing and learning. I agree the optimization convergence rate depends most dominantly on one eigenvalue, but if that is the focus of training, it could lead to some weird biases in the resulting model; it seems a fuller spectral approach could lead to better learning.  However, how one approaches this is probably still an open question, and I admittedly have no idea how to actually do anything like that.
> >
> > Overall, I am happy with the responses and will keep my score.

---

> > > ### Author Response · Authors · 2024-08-10
> > >
> > > We thank the reviewer for their timely and highly constructive response. We highly appreciate the interesting discussion.
> > >
> > > Just to avoid a potential misunderstanding, we would like to highlight that only our propositions are limited to the ring example. In addition, we have provided numerical evidence on real world graphs (e.g. Figure 2) and ER graphs in Figure 3. Our algorithms and insights apply to any real world graph.
> > >
> > > Re Q1: We agree that the full spectral approach could lead to better learning. To understand the intricacies, we would probably also have to consider the full learning dynamics.

---

### Official Review · Reviewer_1tmm · 2024-07-12

**Soundness:** 2
**Presentation:** 3
**Contribution:** 3
**Rating:** 6
**Confidence:** 4

**Summary:**

The paper addresses the issue of over-squashing and over-smoothing in Message Passing Graph Neural Networks (MPNNs). It proposes a novel spectral gap optimization framework with rewiring, inspired by the Braess phenomenon, to mitigate both over-squashing and over-smoothing. The method is computationally efficient and is shown to improve generalization on various benchmarks.

**Strengths:**

1.The idea that “certain edge deletions can maximize the spectral gap” is interesting and makes sense. The proposed framework is described in sufficient detail and easy to follow.

2.The paper is well-written and easy to read.

3.The experimental section involves a wide variety of datasets, providing a comprehensive evaluation of the method's performance.

**Weaknesses:**

1.How universally do the conditions for the inequality mentioned in Eldan's Lemma hold for graph datasets commonly used in the GNN field? Can the authors provide a theoretical or experimental analysis?

2.Why are the results in Sec 4.3 inconsistent between the ER toy graph and other graph datasets? On the ER graph, adding edges leads to a continuous increase in the spectral gap, while deleting edges leads to an initial increase followed by a decrease. However, in common datasets such as Cora, adding edges seems to decrease the spectral gap, while deleting edges increases the spectral gap. Is there a misunderstanding on my part?

3.Does the viewpoint of this paper, "deleting edges to mitigate over-squashing," conflict with some previous rewiring methods that suggest "adding edges to mitigate overfitting"?

4.Why were different baselines chosen for different experiments? For example, DRew was selected for Long Range Graph Benchmark datasets while FoSR was chosen for large heterophilic datasets. What are the insights behind these choices? Additionally, some rewiring methods were not selected as baselines, such as BORF [1].

5.Some typos, ‘basline’ in L187, etc.


[1] Khang Nguyen, et al. “Revisiting Over-smoothing and Over-squashing Using Ollivier-Ricci Curvature” ICML 2023.

**Questions:**

See weaknesses.

**Limitations:**

The authors have addressed the limitations and potential societal impact.

---

> ### Author Rebuttal · Authors · 2024-08-06
>
> We thank reviewer 1tmm for the constructive feedback provided and valuable insights. We include answers for each of the points raised. Tables are located in the document on the global response.
> * W1. The criterion states that if the quantity $g$ is positive, then the Braess paradox occurs. In all of the datasets for which we have tested our EldanDelete method, we have found edges that satisfy this criterion, and we have been able to increase the spectral gap by deleting those edges. The criterion is not necessary for the Braess paradox to occur, and the Braess paradox is not necessarily found in all graphs, but we have been able to apply it to all real-world graphs we have tested -for at least one edge, and most times for many. In the attachment (T3) we include how many edges we can delete iteratively such that all have positive $g$ for the set of heterophilic graphs.
> * W2. The ER graph is a toy example to show that our proposed algorithms indeed help in spectral gap maximization. In Figure 3a, we add edges to the ER graph to show that our proposed EldanAdd and ProxyAdd lead to spectral gap improvements. In Figure 3b we use ProxyDelete and EldanDelete to show that we can delete edges and still increase the spectral gap. However, there might not be many edges that satisfy the criterion, and from a point onward, further deletions will eventually decrease the gap. This is no problem, since we only want to modify a small number of edges without changing the original degree distribution of the graph. For real-world graphs such as Cora, we present in Tables 16 and 17 the increase in spectral gap for both additions and deletions. We also present the values for ProxyDelete, ProxyAdd, and FoSR for other datasets in the attachment (T1). Concluding, our edge rewiring decisions always increase the spectral gap (unless we aim for more extreme graph sparsification when we aim for the smallest decrease that is possible).
> * W3. Our proposed method is not in conflict with the previous literature but complements it. Current consensus for mitigating over-squashing suggests rewiring the graph as a possible solution. Our proposed method aligns with this motivation; more specifically, we show that deleting edges for spectral gap maximization can help mitigate both over-squashing as well as slow down the rate of detrimental over-smoothing, which challenges the commonly held assumption that both of these phenomena are a trade-off. Adding edges is still a powerful tool to solve problems in GNNs related to connectivity and generalization. However, in our work we highlight the advantage of edge deletions (which could also be combined with edge additions), as they could fight over-smoothing in addition to over-squashing.
> * W4. We believe FoSR to be the most comparable method to our own, as it is also a preprocessing spectral rewiring method. Therefore, we included it in all main comparisons. The Long Range Graph Benchmark datasets are conceived as a test to assess if proposed methods can indeed tackle over-squashing, for which we compare with the method that has been proposed for these particular tasks. We appreciate the suggestion to compare it to other rewiring methods. We report the results for FoSR on Long Range Benchmark datasets and also include BORF on other commonly used datasets (as obtained by the authors) in the attachment (T5,6).
> * W5. We thank the reviewer for noticing the typos which we will correct in a revision.

---

> > ### Comment · Reviewer_1tmm · 2024-08-11
> >
> > Thank you for the rebuttal. As my original scoring is optimistic, I retain my scoring for this paper.

---

### Official Review · Reviewer_p1uY · 2024-07-22

**Soundness:** 2
**Presentation:** 3
**Contribution:** 3
**Rating:** 5
**Confidence:** 3

**Summary:**

Inspired by the Braess phenomenon, the paper proposes a Greedy graph pruning algorithm (PROXYDELETE) that maximizes the spectral gap in a computationally efficient way to simultaneously address over-smoothing and over-squashing of GNNs. The paper then verifies the empirical effectiveness of the method on long-range and heterophilic graph tasks, and pruning graphs for lottery tickets

**Strengths:**

1. The paper is clearly written and easy to follow.
2. Evaluation is done on a diverse set of tasks (long range, heterophilic, GLT), showing the empirical effectiveness of the method to certain extent.
3. The method is more principally designed than the common random edge dropping baselines in the literature.
4. The method seems to be efficient in real time.

**Weaknesses:**

1. The theoretical analysis (proposition 3.2-3.4) in Section 3, while it is interesting given the historical context of the Braess’ paradox, is completely based on a specific contrived example in Figure 1. How much the result derived from this example can generalize to general graphs, even a class of general graphs is unclear.

2. The experimental setups are not consistent throughout the paper. The discrepancies that make the claims in the paper seem not reliable in the sense that they might overfit the specific examples:
    - The inspiration is drawn based on the contrived example.
    - To show that the increase in spectral gap can help with the linear ridge regression task, the heterophilic dataset Texas is used; however, there is no clue that the method indeed leads to spectral gap expansion there, nor spectral gap expansion indeed leads to improvement on that data, even we suppose that the method led to spectral gap expansion there.
    - Spectral gap expansion is then shown in Erdos-Renyi graphs in Section 4.3.

These three graphs are three specific graphs (contrived example, Texas and ER graphs). How they are connected to each other and fit together is unclear to me.

3. While the authors show that their Proxy method is better at spectral expansion than Eldan criterion in Section 4.3 on ER graphs, Eldan criterion seems to on par or even outperform Proxy in many setups across all tasks considered in Table 1-5. Wouldn't this contradict the main idea of the paper that "spectral expansion" leads to better performance?

4. I think the paper can be better positioned, too. I don't quite see how the current literature put over-smoothing and over-squashing "diametrically opposed". My understanding is that the over-smoothing theory also have a direct spectral connection. For example, see [1] and [2] for the bounds derived for GCNs and attention-based GNNs respectively, which is directly related to the spectral gap of the graph.


References

1. Oono and Suzuki. Graph neural networks exponentially lose expressive power for node classification.

2. Wu etl al. Demystifying Oversmoothing in Attention-Based Graph Neural Networks.

**Questions:**

1. I don't see how the findings in Section 3 suggests that "deleting edges can address over-squashing and over-smoothing simultaneously." For example, from Fig 2, how could one conclude that "deleting edges helps reduce over-smoothing, while still mitigating over-squashing
via the spectral gap increase." Could you clarify?

2. For clarification, does Lemma 3.1 from Eldan el al. 2017 applies only to Erdos-Renyi graphs or general graphs? As in line 2017 it says that their study is for random graphs.

3. Any justification why for spectral expansion, the proxy method does better than Eldan criterion in Section 4.3?

4. How would the method compare to random edge-dropping baselines such as DropEdge [1] in the literature?

5. I think it would be good to include the real run time of the method as a plus in the main text, as I think this is a main advantage of the method.

References

[1] Rong et al. DropEdge: Towards Deep Graph Convolutional Networks on Node Classification.

---

> ### Author Rebuttal · Authors · 2024-08-06
>
> We thank reviewer p1uY for the constructive feedback provided and valuable insights. We include answers for each of the points raised. Tables are located in the document on the global response.
> * W1. Our theoretical investigations have the purpose of providing a counterexample for the common hypothesis that over-smoothing and over-squashing have to be traded off against each other. It further provides an intuition when the Braess phenomenon can be exploited to alleviate these two problems simultaneously. In experiments, we show that this is a very common phenomenon and can be exploited in all studied real-world graphs. This demonstrates that our insights are clearly application relevant.
> * W2. Our experimental setup is consistent, as the relevant information is available for all datasets. The ring serves as existence proof -for which we obtain provable evidence regarding the spectral gap, over-smoothing, etc.- and, as a simple example, provides an intuition of how both over-squashing and over-smoothing can be mitigated simultaneously -since the current consensus is to view them as a trade-off [3]. For the linear ridge regression setup we have also included analyses for Cora, Citeseer (homophilic), Chameleon and Texas (heterophilic) in Figure 6 of the appendix. The ER graphs merely show that our proposed algorithms indeed lead to spectral gap optimization. In Table 16 and 17 we also present the spectral gap values before and after rewiring for Cora, Citeseer, Chameleon, Squirrel, Roman Empire, Amazon ratings and Minesweeper. As an overview, in the attachment (T1) we report a table of spectral gap changes for ProxyDelete, ProxyAdd, and FoSR on all datasets. The number of deletions varies depending on the size and type of dataset, yet note that the spectral gap is successfully increased (in all cases but Pubmed on deletions).
> * W3. To increase the spectral gap for improved GNN performance has been previously proposed in the over-squashing literature [2,3,4]. Our contribution is to point out that, by deleting edges, one can also achieve this goal, while reducing the over-smoothing rate. Note that spectral gap optimization is a label independent rewiring strategy. As our ring example (and the no free lunch theorem) suggest, no rewiring strategy can be universally optimal. For specific label distributions, Eldan’s criterion can therefore outperform ProxyGap. Yet, as it is similarly based on the Braess paradox, it has the same advantage of simultaneously addressing over-squashing and over-smoothing in principle.
> * W4/Q1. Some theory on over-smoothing draws on an analogy to random walks [3], where the spectral gap controls the convergence speed to the steady state and thus the rate of smoothing. According to this view, one would need to determine the right amount of smoothing: if it is too high, over-smoothing should occur; if it is too little, over-squashing hinders good GNN performance. In contrast, we follow the narrative of [6] that over-smoothing refers to unwanted neighborhood aggregation, which also depends on other aspects (like heterophily structure) and is directly deduced from GNN generalization performance. In our work we have described how this depends on features and node labels, which explains why our proposed method is particularly effective in heterophilic settings. Moreover, we have shown several and diverse real-world datasets where our method (1) reduces the rate of smoothing according to the testbed of [6] (Figures 2, 6), and (2) increases the spectral gap by deleting edges that fulfill the Braess paradox. If over-smoothing and over-squashing were exclusively defined by the graph’s spectrum, both problems could not be decoupled, as has been the current assumption in graph rewiring literature [2,3,5]. We believe that these are relevant theoretical (as well as practical) insights that add a novel perspective to the discussion of graph rewiring.
> * Q2. It is a general condition for finite graphs. As they state in their section 1.3 (Approach): “We first obtain a general sufficient condition [...]  See Lemma 3.2 for details. Next, we specialize this general condition to Erdos-Rényi random graphs…”. We use their Lemma 3.2.
> * Q3. Eldan gives a sufficient condition or a guarantee that the paradox occurs when deleting the specified edge, but it does not give a direct measurement of the magnitude with which the spectral gap increases. This is why the quantity is generally more conservative than the proxy value.
> * Q4. DropEdge [1] proposes to drop a random percentage of edges of the graph during training, which acts as a graph augmentation technique, and they have access to the entire graph during their whole training procedure. Our method however is a pre-processing technique which actually sparsifies the input graph. During optimization the GNN model consumes a sparser, more incomplete version of the graph, thus making these two methods not directly comparable. We believe it to be more comparable to deleting edges randomly as a pre-processing step. We have nonetheless included both in the attachment (T2,3).
> * Q5. We appreciate the suggestion, and we also agree that efficiency is a big advantage of our method -which we included in Table 14. We would be happy to move it to the main text given sufficient space.
>
> References:
> * [1] Rong et al. DropEdge: Towards deep graph convolutional networks on node classification. ICLR 2020
> * [2] Karhadkar et al. FoSR: First-order spectral rewiring for addressing oversquashing in GNNs. ICLR 2023
> * [3] Giraldo et al. On the Trade-off between Over-smoothing and Over-squashing in Deep Graph Neural Networks. ACM CIKM 2023
> * [4] Banerjee et al. Oversquashing in GNNs through the lens of information contraction and graph expansion. AACCCC 2022
> * [5] Nguyen et al. Revisiting over-smoothing and over-squashing using Ollivier-Ricci curvature. ICML 2023
> * [6] Keriven, N. Not too little, not too much: a theoretical analysis of graph (over)smoothing. NeurIPS 2022

---

> ### Comment · Reviewer_p1uY · 2024-08-12
> **Thank you**
>
> I would like to thank the authors for their detailed response. The rebuttal has addressed my major concerns. One minor thing is that I still think the ring example given is quite specific and the intuition there might not generalize to real graphs. Despite this weakness on theoretical side, the authors have shown the effectiveness of the idea in practice. I will increase my score to 5.

---

### Author Rebuttal · Authors · 2024-08-07

We sincerely thank all reviewers for their feedback. We appreciate the overall positive reception of our work and are glad that its novelty, clarity, and thoroughness have been recognized. Below, we summarize our key contributions and address the added material in the attached document:

* Our work challenges the common hypothesis that over-smoothing and over-squashing must be traded off against each other in graph rewiring. We demonstrate how the Braess paradox can be exploited to alleviate both problems simultaneously through strategic edge deletions.
* We follow the narrative of [6] that over-smoothing refers to unwanted neighborhood aggregation, which depends on features and node labels. This explains our method's effectiveness in heterophilic settings.
* Our theory for the ring graph provides an existence proof that over-smoothing and over-squashing can be optimized jointly and provides insights into the basic mechanism that explains the performance gains by our proposed rewiring methods. We have extended our analyses to real-world graphs (Cora, Citeseer, Chameleon, Texas, etc.) to demonstrate its practical applicability.
* We propose efficient algorithms for spectral gap optimization, which have been shown to be effective across various graph types and tasks.

To address specific requests and provide further clarity, we have included the following additional analyses in the attachment:

* T1: Table of spectral gap changes for ProxyDelete, FoSR, ProxyAdd and EldanDelete on all datasets
* T2, T3: Comparison with random edge deletion and DropEdge
* T4: Number of edges that satisfy the Eldan criterion
* T5, T6: Results for FoSR on Long Range Benchmark datasets, and BORF on commonly used datasets

We believe that our insights and our proposed practical algorithms are timely and of high interest to the community. In particular, our contribution exploits an overlooked piece in the puzzle of how to improve the performance of graph neural networks and/or their computational efficiency by graph rewiring.

---

### Decision · Program_Chairs · 2024-09-25

**Decision:**

Accept (poster)

**Comment:**

The article proposes a graph pruning algorithm that maximises the spectral gap in a computationally efficient way to address oversquashing and oversmoothing in GNNs. It observes that, by deleting edges, one can increase the spectral gap, while reducing the oversmoothing rate.

Reviewers find that the paper is clearly written and easy to follow, that the evaluation uses a diverse set of tasks, and the method appears to be efficient. They regarded it as a weakness that the theoretical analysis was based on a specific example and thus unclear how it would generalize, the experimental setups not always consistent, and connections to the lottery ticket hypothesis not sufficiently motivated. The rebuttal addressed several of the main concerns, prompting some to increase the score and others to maintain their positive score.
At the end of the discussion period, all reviewers recommended weak accept.

I conclude that the article provides a well presented and meaningful advance on a topic of interest and therefore recommend accept. The authors are asked to implement the promised revisions in the final version of the article.